# Rheological Model and Transition Velocity Equation of a Polymer Solution in a Partial Pressure Tool

**DOI:** 10.3390/polym11050855

**Published:** 2019-05-10

**Authors:** Bin Huang, Xinyu Hu, Cheng Fu, Chongjiang Liu, Ying Wang, Xu An

**Affiliations:** 1Key Laboratory of Enhanced Oil Recovery (Northeast Petroleum University), Ministry of Education, College of Petroleum Engineering, Northeast Petroleum University, Daqing 163318, China; 15204593438@163.com; 2Post-Doctoral Scientific Research Station, Daqing Oilfield Company, Daqing 163413, China; 3Research Institute of Production Engineering, Daqing Oilfield, Daqing 163453, China; liuchongjiang@petrochina.com.cn; 4Aramco Asia, Beijing 100102, China; ying.wang.1@aramcoasia.com; 5NO.2 Oil Production Company, Daqing Oilfield, Daqing 163453, China; anxu212@126.com

**Keywords:** partial pressure tool, rheological equation, transition velocity, pressure drop, apparent viscosity

## Abstract

In order to solve the problem of the low production degree of oil layers caused by an excessively large permeability difference between layers during polymer flooding, we propose partial pressure injection technology using a partial pressure tool. The partial pressure tool controls the injection pressure of a polymer solution through a throttling effect to improve the oil displacement effect in high- and low-permeability reservoirs. In order to analyze the influence of the partial pressure tool on the rheological property of the polymer solution, a physical model of the tool is established, the rheological equation of the polymer solution in the partial pressure tool is established according to force balance analysis, the transition velocity equation for the polymer solution is established based on the concept of stability factor, and the influence of varying the structural parameters of the partial pressure tool on the rheological property of the polymer solution is analyzed. The results show that the pressure drop of the polymer solution increases with the decrease of the front groove angle of the partial pressure tool (from 60° to 30°), reaching a maximum of 1.77 MPa at a front groove angle of 30°. Additionally, the pressure drop of the polymer solution increases with the decrease of the outer cylinder radius (from 25 to 24 mm), reaching a maximum of 1.32 MPa at a radius of 24 mm. However, the apparent viscosity of the polymer solution before and after flowing through the partial pressure tool does not change for any of the studied parameters. These research results are of great significance to research on partial pressure injection technology and enhanced oil recovery.

## 1. Introduction

With the development of oilfields, many alternatives have been taken to improve their production efficiency. The process of polymer flooding in oilfields can improve the injection profile to a certain extent. However, due to the influence of various factors, such as reservoir heterogeneity and well-pattern perfection, interlayer contradiction has not been fundamentally solved. Under the general injection mode, high-molecular-weight polymer enters more into high-permeability reservoirs and less into thin and poor reservoirs, which greatly reduces the permeability and production degree of thin and poor reservoirs [1,2,3]. However, the direct injection of low-molecular-weight polymer solution can increase the degree of control of polymer flooding, although the overall flooding effect will be affected to some extent as the flooding effect of the high-permeability layer decreases. In order to improve the overall development effect of polymer flooding and improve interlayer contradiction, Daqing Oilfield has proposed a partial pressure polymer injection process. The main technical idea is to use a partial pressure tool to reduce the injection pressure of polymer solution while ensuring the oil displacement effect of high-molecular-weight polymer in the high-permeability layer, so as to control the injection amount in the high-permeability layer, make the polymer solution enter the low-permeability layer to produce oil displacement, and improve the overall development effect.

Meyers [4] carried out polymer flooding experiments in two newly developed wells by means of chemically enhanced oil recovery in Kottrabathe oilfield. The results show that the oil recovery efficiency of polymer flooding in the total affected area was 60.6% and the oil recovery efficiency of water flooding in the total affected area was 39.9%. Clark [5] conducted polymer flooding research in the West Kiehl oilfield, and conducted oil displacement tests by means of mobility control, profile modification, and the injection of combined mobility control profile. The experiments showed that polymer flooding successfully displaced residual oil that could not be displaced during water flooding, and the oil recovery ratio increased from 11% to 40%. An oil displacement experiment was carried out by injecting a high-concentration polymer solution into heterogeneous sediment cores, Denney [6] concluded from experiments that when the pressure gradient and interfacial tension of oil layers remain unchanged, polymer flooding can achieve a production efficiency that is 23.87% higher than that of water flooding under the same conditions. Using simulation and core flooding for enhanced oil recovery screening, Wassmuth et al. [7] proposed that in heavy oil reservoirs, polymer flooding is the preferred enhanced oil recovery technology after primary oil recovery and water flooding. Additionally, when the water cut reaches 95% after water injection, polymer flooding can improve the oil recovery ratio by 20%.

Some scholars have concluded that polymer flooding can only expand the volume of the flood, and cannot improve the efficiency of oil displacement, so that the oil recovery value is lower [8]. The reason for this conclusion was that they considered the polymer solution as a Newtonian fluid [9,10]. However, in fact, polymer solutions are non-Newtonian fluids. Enhanced oil recovery is mainly allowed by mechanisms which increase the viscosity of, and reduce the permeability of, the water phase, which reduces the fluidity of the water phase and the fluidity ratio of water flooding, and thus improves the sweep efficiency of the flooding and achieves the purpose of enhanced oil recovery. Additionally, due to its rheological properties, the polymer exerts a stretching effect on the oil film or oil droplets in the flow process, which increases the carrying force and improves the micro-oil washing efficiency of the polymer solution. Therefore, it is of great significance to study the rheology of polymer solutions for the polymer flooding injection process.

Yin et al. [11] simulated the flow characteristics of polymer solutions at both ends of the pores in a reservoir by means of numerical simulation, drew contour lines of polymer solution velocity and flow function, and quantitatively calculated the microscale scanning efficiency of the solutions. The simulation results show that the sweep area and displacement efficiency increase with the increase of the viscoelasticity of the polymer solution. Compared with Newtonian fluids, the viscoelasticity of polymer solutions can improve the displacement efficiency of pores. By establishing rheological equations, Lee [12] elaborated that the rheological property of a polymer solution for enhanced oil recovery depends on the molecular properties, concentration, salinity, shear rate, and temperature of the polymer. Rheological measurements were carried out using commonly used polymers, and rheological model parameters related to these variables were determined. The results show that with increasing temperature, salinity, and shear rate, the concentration and viscosity of the polymers decreased. At the same time, the oscillatory rheological characteristics of the polymers were measured in order to determine the flow behavior of the polymers in a reservoir rock. Silva [13] studied the rheological behavior of different polymer solutions in porous media through laboratory tests of fluid flow in porous media and analysis of fluid rheology and the hydrodynamic diameters of polymer molecules. Due to the increasing hydrodynamic volume of the polymer during the shearing and dilution of the polymer solution, the adsorption of the polymer solution was increased, indicating that polymer flooding can improve oil recovery in oilfields.

The above mentioned experiments of polymer flooding and the rheology of polymer solutions show that polymer flooding can improve the oil recovery ratio, and that the flooding effect is higher than that of water flooding. However, the relationship between the permeability of different oil layers is not considered in the process of polymer injection. High-molecular-weight polymer solutions have better oil displacement effects in high-permeability oil layers; however, they have poor oil displacement effects in low-permeability oil layers and limited degrees of enhanced oil recovery. Therefore, in order to improve such interlayer contradictions, Li et al. [14] proposed separate polymer flooding injection technology including concentric separate injection technology, eccentric separate injection technology, and stratified separate injection technology according to the development needs of different stages and different displacement objects in the Daqing Oilfield in order to adjust the polymer injection pressure and control the polymer injection volume. The experimental results show that the recovery ratio in the central well was increased by 5.08% on the basis of the original polymer flooding effect. Additionally, based on a separate polymer injection technology, Geng et al. [15] put forward intermittent injection, annular depressurization injection, and concentric injection of slender tubes. The pressure drop of the polymer solution reached 2.4 MPa and the control range of the polymer solution injection volume was greatly improved.

Although partial pressure injection technology and matching tools are proposed, which can adjust the injection pressure of polymer solution and improve the injection relationship between oil layers, these methods, while adjusting the pressure, cause a large reduction in viscosity when the polymer solution is injected into a high-permeability oil layer, due to shearing action. When a high pressure drop occurs, the viscosity reduction can be as high as 40% [15], which seriously affects the oil displacement effect of the polymer in high- and low-permeability oil layers, preventing a maximum oil recovery rate. Therefore, it is necessary to establish a partial pressure tool that can generate a large drop in the pressure of the polymer solution while minimizing the viscosity reduction.

The analysis of the rheological characteristics of non-Newtonian fluids, such as polymer solutions, in specific tools, and the method of establishing the equation, can be used to deduce the rheological equation of a polymer solution in partial pressure tools, which is helpful for analyzing the influence of partial pressure tools on pressure drops in, and the apparent viscosity of, a polymer solution. Wu et al. [16] used the White–Metzner constitutive model to describe the rheological properties of a polymer solution. A finite-element method was used to numerically solve the flow of the polymer solution through a single transverse groove, the pore pressure under different Deborah numbers was obtained, and the influence of fluid characteristic parameters and groove geometry on pore pressure was analyzed. Huang et al. [17] established the rheological equation of a polymer solution in a mass separation tool according to the tensor form of the rheological equation and the local perturbation theory, and deduced the formula of the optimal injection speed of the polymer solution. Malkus [18] numerically solved the flow of a polymer fluid through a transverse groove by an improved nonlinear iterative method, and analyzed the relationship between the first normal stress difference and the pore pressure. Wang [19] established the governing equation for the unsteady flow of a non-Newtonian fluid in the eccentric annuli of an inner tube. The calculation formula for the pressure distribution on the wall of the inner tube was given. Taking a polymer solution as an example, the governing equation and the calculation formula above equations were numerically solved and calculated by the finite difference method.

Hidema [20] studied the relationship between the tensile rheological properties of polymer solutions and the turbulent deformation of the solutions, and determined that there are three kinds of flow states in the two-dimensional turbulent flow of a polymer solution, and that the vortex shedding in the two-dimensional flow is divided into three types, which are affected by the relaxation time of the polymer solution. Furthermore, research by Wouter [21] showed that unsteady turbulence of polymer solutions unsteady turbulence shows general characteristics before reaching a dynamic equilibrium state. The nonequilibrium correction of the Kolmogorov spectrum can explain the observed universal nonequilibrium scale, and can be used to establish a wide range of unsteady turbulence models.

As the above research shows, the technologies related to polymer flooding are very mature; however, there are few researches related to improving interlayer contradiction, controlling the injection amount of polymer solution in high- and low-permeability oil layers, and further improving the oil displacement effect of polymer flooding. Only the Daqing Oilfield has proposed the concept of partial pressure polymer solution injection technology to solve this problem. Existing theories and tools can be used to adjust the pressure of polymer solutions; however, they produce large reductions in the viscosity of the solutions and affect their oil displacement effect. Additionally, the rheological properties of polymer solutions in partial pressure tools have not yet been studied. The establishment of mathematical models for the rheological properties of polymer solutions in partial pressure tools is relatively single sample, lacks theoretical basis, and does not consider the problem of fluid state transition. Therefore, the establishment of rheological models of partial pressure tools that produce large pressure drops and low viscosity reductions of polymer solutions is extremely important for enhanced oil recovery in oilfields.

In this paper, a physical model of a partial pressure tool is established, and the rheological equation of a polymer solutions in an annular flow channel and throttling section of the partial pressure tool is deduced according to stress balance analysis. According to the theory of turbulence and dissipation effect, and through numerical simulation of the polymer solution in the partial pressure tool, the concept of transition velocity is proposed, and the transition velocity equation is deduced based on the concept of stability factor. According to the production situation of the oilfield, a polymer solution with a molecular weight of 1600 × 10^4^ and a concentration of 1000 mg/L is prepared, and shear testing and numerical fitting are carried out. Relevant parameters are substituted into the rheological equation in order to obtain the pressure and apparent viscosity values of the polymer solution under different structural parameters of the partial pressure tool. The degree of influence of the partial pressure tool on the pressure and apparent viscosity of the polymer solution with different structural parameters is analyzed, and the optimal structural parameters are obtained. Moreover, the transition velocity of the polymer solution in the partial pressure tool is calculated in order to judge the flow state.

## 2. Methodology

### 2.1. Physical Experiments

Experimental chemicals:

The polymer used to prepare the polymer solution in this experiment was partially hydrolyzed polyacrylamide (HPAM) with a relative molecular mass of 1600 × 10^4^, a degree of hydrolysis of 26%, and a mass fraction of 90.17%. The HPAM was provided by the Daqing Oilfield Production Technology Research Institute. The experimental water used for preparing the polymer solution was provided by the No. 1 Oil Production Plant of the Daqing Oilfield; a filter membrane was required before use. The composition of the experimental water is shown in Table 1.

Experimental instruments:

YP-B2003 electronic balance (Huaguang instrument factory, Liaoyang, China); EURO-ST D S25 agitator (IKA, Staufen, Germany); RS-150 rheometer (IKA, Staufen, Germany).

Experimental method: The YP-B2003 electronic balance was used to measure dry powder of the HPAM and experimental water in proportion to prepare a polymer solution with a concentration of 1000 mg/L;The polymer mother liquor was dissolved and agitated using the EURO-ST D S25 electronic agitator at 250 r/min for 2.5 h and then left to stand for 2 h to ensure full dispersion of the solution molecules and a uniform system;The thermostatic circulation system was started in the RS-150 rheometer, heated to 45 °C, and put in homeostasis for 15 min. The prepared solution was put into a preheated measuring outer cylinder, and the temperature was kept constant for 20 min so that the temperature of each point of the sample could reach the testing temperature;The shear rate was set from 1 to 1000 s^−1^, the rheometer was started, and the rheometer’s viscosity option was selected for testing. When the indication viscosity value was basically stable, recording was started, and then recording was performed every 5 min. Four viscosity values were recorded continuously, and if the deviations between the first value and the three other values did not exceed 5%, the system was considered to have reached dynamic equilibrium.

### 2.2. Laminar Flow Model of the Polymer Solution in the Partial Pressure Tool

#### 2.2.1. Physical Model of the Partial Pressure Tool

In order to facilitate the application and popularization of polymer flooding partial pressure injection technology in oilfield production, it is of great significance to study the partial pressure tool model and the rheological characteristics of polymer solutions in the partial pressure tool. First, a physical model of the partial pressure tool is established for this research. Figure 1 and Figure 2 show a three-dimensional and a two-dimensional representation of the partial pressure tool model, respectively.

The working principle of the partial pressure tool is to reduce the pressure of the polymer solution by changing the flow area. Therefore, a multistage model with a variable cross-section was designed to produce a small throttling loss of the polymer solution under a section of a throttling unit. In this way, multiple throttling units are combined together, and the final cumulative value of polymer solution pressure drop meets the throttling loss. The throttling section forms a flow channel with the inner wall of the outer cylinder. In order to reduce the shear stress of the polymer solution at the minimum cross-section area, the convex part of the throttling section is designed in a circular arc shape, and in order to make the solution flow evenly in the partial pressure tool, the partial pressure tool is designed axisymmetrically. An annular flow channel is included at the front and back of the throttling section so that the solution is in a fully developed section in the inlet section and the outlet section.

#### 2.2.2. Flow Model of the Polymer Solution in the Annular Flow Channel of the Partial Pressure Tool

The polymer solution used in this study is a non-Newtonian fluid. Furthermore, experiments on the effect of polymer degradation on polymer flooding in heterogeneous reservoirs [22] and the effect of mechanical degradation of the polymer solution on polymer injection performance in porous media [23] have proved that polymer solutions are power-law fluids in actual situations.

Moreover, the designed partial pressure tool is multistage, and the degree of the shearing effect on the polymer solution is limited. In the range of controlled shear rates, the polymer solution in the process of passing through the partial pressure tool is considered to be a power-law fluid.

According to the shear test, the relationship curve between polymer solution viscosity and shear rate are plotted in Figure 3.

Figure 3 shows that the polymer solution conforms to the flow characteristics of a power-law fluid. That is, with increasing shear rate, the viscosity decreases. Thus, the polymer solution is a power law fluid.

Constitutive Equation of Polymer Solution: (1)$$f(\tau )=\dot{\gamma }={\left(\frac{\tau }{k}\right)}^{\frac{1}{n}}$$
where *τ* is the shear stress, in Pa; *k* is the consistency coefficient to characterize the viscosity of the polymer solution, in Pa·s; *n* is the rheological index of the solution, indicating the degree to which the flow characteristics deviate from a Newtonian fluid; and $\dot{\gamma }$ is the average strain rate, in s^−1^.

Before the polymer solution enters the throttling section of the partial pressure tool, the solution flows in the annular flow channel, as shown in Figure 4. It is assumed that the flow of the polymer solution always conforms to the flow characteristics of a power-law fluid, that the flow state is laminar, that the temperature of the polymer solution does not change during the flow, and that the effect of gravity on the solution is nil.

Annular flow channel of the partial pressure tool typical sizes are showed in Table 2.

According to the balance between pressure and shear stress:(2)$$\Delta p\pi \left\{{\left[\frac{1}{2}(R_{o}+R_{i})+r\right]}^{2}-{\left[\frac{1}{2}(R_{o}+R_{i})-r\right]}^{2}\right\}=2\pi \left\{\left[\frac{1}{2}(R_{o}+R_{i})+r\right]+\left[\frac{1}{2}(R_{o}+R_{i})-r\right]\right\}L\tau$$
(3)$$\tau =\frac{\Delta pr}{L}$$
where Δp is the pressure drop in the polymer solution, in Pa.

At the outer wall of the annular flow channel:(4)$$\tau_{\omega }=\frac{\Delta p(R_{o}-R_{i})}{2L}$$

It is assumed that the flow velocity of the polymer solution in the partial pressure tool conforms to simple shear flow, that the shear rate is the same as the velocity gradient, and that the flow direction is the opposite, that is:(5)$$\dot{\gamma }=-\frac{du}{dr}$$
(6)$$u=\int_{0}^{\frac{1}{2}(R_{o}-R_{i})}f(\tau )dr$$
where u is the velocity of the polymer solution, in m/s.

Integral variable substitution is as follows:(7)$$u=\frac{R_{o}-R_{i}}{2\tau_{\omega }}\int_{\tau }^{\tau_{\omega }}f(\tau )d\tau$$
(8)$$u=\frac{n}{n+1}{\left(\frac{\Delta p}{kL}\right)}^{\frac{1}{n}}\left\{{\left[\frac{1}{2}(R_{o}-R_{i})\right]}^{\frac{n+1}{n}}-r^{\frac{n+1}{n}}\right\}$$

Flow rate is expressed in integral form thus:(9)$$dQ=u\times 2\pi \left[\frac{1}{2}(R_{o}+R_{i})+r\right]dr+u\times 2\pi \left[\frac{1}{2}(R_{o}+R_{i})-r\right]dr$$
(10)$$Q=2\pi (R_{o}+R_{i})\int_{0}^{\frac{1}{2}(R_{o}-R_{i})}udr$$
where *Q* is flow rate, in m/s^3^.

Integrating by parts Equation (10), when $r=\frac{1}{2}(R_{o}-R_{i})$, u=0:(11)$$Q=\pi (R_{o}^{2}-R_{i}^{2})(\frac{n}{2n+1}){\left(\frac{\Delta p}{kl}\right)}^{\frac{1}{n}}{\left(\frac{R_{o}-R_{i}}{2}\right)}^{\frac{n+1}{n}}$$

Equation (11) converts variables in order to obtain the analytical equation of pressure drop: (12)$$\Delta p={\left[\frac{2Q}{\pi (R_{o}^{2}-R_{i}^{2})(R_{o}-R_{i})}\frac{2n+1}{n}\right]}^{n}\frac{2kl}{R_{o}-R_{i}}$$

The average shear rate is: (13)$$\bar{\dot{\gamma }}=\int_{0}^{\frac{1}{2}(R_{o}-R_{i})}\frac{2\dot{\gamma }}{R_{o}-R_{i}}dr$$

Putting Equation (3) into Equation (1), the shear rate is:(14)$$\dot{\gamma }={\left(\frac{\Delta p}{kl}\right)}^{\frac{1}{n}}r^{\frac{1}{n}}$$

The average shear rate is integrated and then put into the analytic equation of the flow rate: (15)$$\dot{\gamma }=\frac{2Q}{\pi (R_{o}^{2}-R_{i}^{2})(R_{o}-R_{i})}\frac{2n+1}{n+1}$$

Based on $\dot{\tau }=k{\left(\bar{\dot{\gamma }}\right)}^{n}$, $\dot{\tau }=\bar{\mu }\bar{\dot{\gamma }}$, the average viscosity of a polymer solution is:(16)$$\bar{\mu }=k{\left(\bar{\dot{\gamma }}\right)}^{n-1}$$
where $\bar{\mu }$ is the average viscosity, in Pa·s.

Putting Equation (16) into Equation (15) gives:(17)$$\bar{\mu }=k{\left[\frac{2Q}{\pi (R_{o}^{2}-R_{i}^{2})(R_{o}-R_{i})}\frac{2n+1}{n+1}\right]}^{n-1}$$

#### 2.2.3. Flow Model of the Polymer Solution in the Throttling Section of the Partial Pressure Tool

When the polymer solution flows into the throttling section from the annular flow channel, as shown in Figure 5, the flow state is consistent with that in the annular flow channel. However, the flow of the polymer solution in the throttling section is a variable cross-sectional flow. In the process of solution flow, the flow cross-section changes periodically from a state of contraction to a state of expansion. In this process, the polymer solution produces differential throttling pressure.

Typical sizes of the throttling section of the partial pressure tool are shown in Table 3.

The pressure drop Equation (13) and the viscosity Equation (18) of the annulus flow channel in the laminar flow state are changed to the equation of the variable cross section:(18)$$\Delta p={\left[\frac{2Q}{\pi (R_{o}^{2}-a^{2})(R_{o}-a)}\frac{2n+1}{n}\right]}^{n}\frac{2kl}{R_{o}-a}$$
(19)$$\bar{\mu }=k{\left[\frac{2Q}{\pi (R_{o}^{2}-a^{2})(R_{o}-a)}\frac{2n+1}{n+1}\right]}^{n-1}$$
where a is the variable cross-sectional inner diameter, in m; m is the number of throttling sections (0<m≤8); x is the length of the partial pressure tool through which the polymer solution flows, in m.

When the polymer solution flows through different sections of the throttling section of the partial pressure tool, the solution’s pressure drops and its viscosity changes, and the modified pressure drop and viscosity equations are: When $x\leq l_{1}$, the polymer solution flows in the annulus flow channel:(20)$$a=R_{i}$$
(21)$$\Delta p={\left[\frac{2Q}{\pi (R_{o}^{2}-R_{i}^{2})(R_{o}-R_{i})}\frac{2n+1}{n}\right]}^{n}\frac{2kl}{R_{o}-R_{i}}$$
(22)$$\bar{\mu }=k{\left[\frac{2Q}{\pi (R_{o}^{2}-a^{2})(R_{o}-a)}\frac{2n+1}{n+1}\right]}^{n-1}$$When $l_{1}+(m-1)c<x\leq (m-1)c+l_{1}+l_{2}$, the polymer solution flows into the throttling section and flows through the contraction part of the throttling section:(23)$$a=\tan\theta \left[x-l_{1}-(m-1)c\right]+R_{i}$$
(24)$$\Delta p={\left[\frac{2Q{\left(R_{o}-\tan\theta \left[x-l_{1}-(m-1)c\right]-R_{i}\right)}^{-1}}{\pi (R_{o}^{2}-{\left\{\tan\theta \left[x-l_{1}-(m-1)c\right]+R_{i}\right\}}^{2})}\frac{2n+1}{n}\right]}^{n}\frac{2kl}{R_{o}-\tan\theta \left[x-l_{1}-(m-1)c\right]-R_{i}}$$
(25)$$\bar{\mu }=k{\left[\frac{2Q{\left(R_{o}-\tan\theta \left[x-l_{1}-(m-1)c\right]-R_{i}\right)}^{-1}}{\pi (R_{o}^{2}-{\left\{\tan\theta \left[x-l_{1}-(m-1)c\right]+R_{i}\right\}}^{2})}\frac{2n+1}{n+1}\right]}^{n-1}$$When $l_{1}+l_{2}+(m-1)c<x\leq (m-1)c+l_{1}+l_{2}+l_{3}$, the polymer solution flows into the throttling section and flows through the middle arc contraction part of the throttling section:(26)$$a=\sqrt{{r_{o}}^{2}-\left\{\sin\theta r_{o}-{\left[x-l_{1}-l_{2}-(m-1)c\right]}^{2}\right\}}+\tan\theta l_{2}-\cos\theta r_{o}+R_{i}$$
(27)$$\begin{array}{l} \Delta p={\left[\frac{2Q{\left(R_{o}-\left\{\sqrt{{r_{o}}^{2}-\left\{\sin\theta r_{o}-{\left[x-l_{1}-l_{2}-(m-1)c\right]}^{2}\right\}}+\tan\theta l_{2}-\cos\theta r_{o}+R_{i}\right\}\right)}^{-1}}{\pi (R_{o}^{2}-{\left\{\sqrt{{r_{o}}^{2}-\left\{\sin\theta r-{\left[x-l_{1}-l_{2}-(m-1)c\right]}^{2}\right\}}+\tan\theta l_{2}-\cos\theta r_{o}+R_{i}\right\}}^{2})}\frac{2n+1}{n}\right]}^{n}\\ \times \frac{2kl}{R_{o}-\left\{\sqrt{{r_{o}}^{2}-\left\{\sin\theta r_{o}-{\left[x-l_{1}-l_{2}-(m-1)c\right]}^{2}\right\}}+\tan\theta l_{2}-\cos\theta r_{o}+R_{i}\right\}} \end{array}$$
(28)$$\bar{\mu }=k{\left[\frac{2Q{\left(R_{o}-\left\{\sqrt{{r_{o}}^{2}-\left\{\sin\theta r_{o}-{\left[x-l_{1}-l_{2}-(m-1)c\right]}^{2}\right\}}+\tan\theta l_{2}-\cos\theta r_{o}+R_{i}\right\}\right)}^{-1}}{\pi (R_{o}^{2}-{\left\{\sqrt{{r_{o}}^{2}-\left\{\sin\theta r_{o}-{\left[x-l_{1}-l_{2}-(m-1)c\right]}^{2}\right\}}+\tan\theta l_{2}-\cos\theta r_{o}+R_{i}\right\}}^{2})}\frac{2n+1}{n+1}\right]}^{n-1}$$When $l_{1}+l_{2}+l_{3}+(m-1)c<x\leq (m-1)c+l_{1}+l_{2}+l_{3}+l_{4}$, the polymer solution flows into the throttling section and flows through the middle arc expansion part of the throttling section
(29)$$a=\sqrt{{r_{o}}^{2}-{\left[x-l_{1}-l_{2}-l_{3}-(m-1)c\right]}^{2}}+\tan\theta l_{2}-\cos\theta r_{o}+R_{i}$$
(30)$$\begin{array}{l} \Delta p={\left[\frac{2Q{\left(R_{o}-\left\{\sqrt{{r_{o}}^{2}-{\left[x-l_{1}-l_{2}-l_{3}-(m-1)c\right]}^{2}}+\tan\theta l_{2}-\cos\theta r_{o}+R_{i}\right\}\right)}^{-1}}{\pi (R_{o}^{2}-{\left\{\sqrt{r_{o}^{2}-{\left[x-l_{1}-l_{2}-l_{3}-(m-1)c\right]}^{2}}+\tan\theta l_{2}-\cos\theta r_{o}+R_{i}\right\}}^{2})}\frac{2n+1}{n}\right]}^{n}\\ \times \frac{2kl}{R_{o}-\left\{\sqrt{{r_{o}}^{2}-{\left[x-l_{1}-l_{2}-l_{3}-(m-1)c\right]}^{2}}+\tan\theta l_{2}-\cos\theta r_{o}+R_{i}\right\}} \end{array}$$
(31)$$\bar{\mu }=k{\left[\frac{2Q{\left(R_{o}-\left\{\sqrt{r^{2}-{\left[x-l_{1}-l_{2}-l_{3}-(m-1)c\right]}^{2}}+\tan\theta l_{2}-\cos\theta r_{o}+R_{i}\right\}\right)}^{-1}}{\pi (R_{o}^{2}-{\left\{\sqrt{r^{2}-{\left[x-l_{1}-l_{2}-l_{3}-(m-1)c\right]}^{2}}+\tan\theta l_{2}-\cos\theta r_{o}+R_{i}\right\}}^{2})}\frac{2n+1}{n+1}\right]}^{n-1}$$When $l_{1}+l_{2}+l_{3}+l_{4}+(m-1)c<x\leq mc+l_{1}$, the polymer solution flows into the throttling section and flows through the contraction part of the throttling section
(32)$$a=\left\{c-\left[x-(m-1)c-l_{1}\right]\right\}\tan\alpha +R_{i}$$
(33)$$\begin{array}{l} \Delta p={\left[\frac{2Q{\left(R_{o}-\left\{c-\left[x-(m-1)c-l_{1}\right]\right\}\tan\alpha -R_{i}\right)}^{-1}}{\pi (R_{o}^{2}-{\left\{\left\{c-\left[x-(m-1)c-l_{1}\right]\right\}\tan\alpha +R_{i}\right\}}^{2})}\frac{2n+1}{n}\right]}^{n}\\ \times \frac{2kl}{R_{o}-\left\{\left\{c-\left[x-(m-1)c-l_{1}\right]\right\}\tan\alpha +R_{i}\right\}} \end{array}$$
(34)$$\bar{\mu }=k{\left[\frac{2Q{\left(R_{o}-\left\{\left\{c-\left[x-(m-1)c-l_{1}\right]\right\}\tan\alpha +R_{i}\right\}\right)}^{-1}}{\pi (R_{o}^{2}-{\left\{\left\{c-\left[x-(m-1)c-l_{1}\right]\right\}\tan\alpha +R_{i}\right\}}^{2})}\frac{2n+1}{n+1}\right]}^{n-1}$$

The function of the partial pressure tool is to reduce the pressure of the polymer solution and to ensure that the apparent viscosity reduction of the polymer solution during the flow is small. From the derived pressure drop and apparent viscosity equations, it can be seen that the pressure drop and apparent viscosity are related to the properties of the polymer solution and the mechanical structure of the partial pressure tool. That is, when different polymer solutions flow through a partial pressure tool with different structural parameters, the pressure drop and apparent viscosity change are different.

#### 2.2.4. Transition Velocity of the Polymer Solution in the Partial Pressure Tool

As the polymer solution flows in a laminar flow state in the partial pressure tool, the flow velocity changes when the solution flows through the variable cross-section of the throttling tool. When the velocity reaches a certain value, the degree of perturbation of the flow increases and the flow state changes from laminar to turbulent, as is consistent with the theory of turbulent flow [24] and the formation law of non-Newtonian fluid turbulence [25].

In order to more clearly observe the change in the flow state of the polymer solution in the partial pressure tool from laminar flow to turbulent flow, as well as the influence of the change in flow state on the pressure of the solution, the Fluent software(ANSYS, Pittsburgh, PA, USA) was used for numerical simulation. During the simulation process, the model structure was not changed, while the flow rate was gradually increased, thus obtaining a vectorgraph and a pressure nephogram of the polymer solution under both a laminar flow and a turbulent flow state. Figure 6 is the laminar flow vectorgraph of the polymer solution in the partial pressure tool, Figure 7 is the turbulent flow vectorgraph of polymer solution in the partial pressure tool, Figure 8 is the laminar flow pressure nephogram of the polymer solution in the partial pressure tool, and Figure 9 is the turbulent flow pressure nephogram of polymer solution in the partial pressure tool.

Figure 6 shows that when the polymer solution flows through the throttling section, although the flow direction changes, each layer of fluid maintains its own flow state, does not interfere with the other layers, and maintains a laminar flow state. However, as shown in Figure 7, when the velocity of the polymer solution increases, the fluid layers cannot maintain their respective flow states, the laminar flow state is destroyed, and there is sliding and mixing between the adjacent flow layers, eventually forming turbulent flow. The degree of perturbation of the polymer solution is greatest at the lowest point of the flow channel. By considering the dissipation effect [26,27] and the turbulent perturbation theory [28], it is concluded that the turbulence intensity generated by the flow disturbance and the energy loss of the solution are largest in the partial pressure tool.

Figure 8 and Figure 9 are pressure nephograms of the polymer solution flowing in the partial pressure tool under laminar and turbulent states. Under the laminar flow state, the pressure change of the polymer solution is relatively uniform. Under the turbulent flow state, at the place in the tool where the perturbation zone is generated, the pressure suddenly becomes smaller upstream of the perturbation zone and becomes larger downstream of the perturbation zone, causing an uneven change in the pressure of the polymer solution, which causes the overall pressure in the tool to drop. This is consistent with the analysis of the dissipation effect described above. The development of turbulence accelerates the energy consumption of the polymer solution compared with the laminar flow. The pressure drop is larger than the laminar flow, so it is very important to study the transition from laminar to turbulent flow of the polymer solution in the partial pressure tool.

For Newtonian fluids, the critical Reynolds number ${Re}_{c}$ is the criterion for judging the flow state; when its value reaches 2100, the flow changes from laminar to turbulent. For non-Newtonian fluids, the critical Reynolds number signifying the change from laminar to turbulent flow is different for different rheological indices. Studies have found that the stability parameter Z can distinguish the flow state of non-Newtonian fluids [29,30]. When Z is equal to 808, it is considered that the fluid changes from laminar to turbulent flow. Therefore, in the present study, the stability parameter Z is adopted as the criterion for judging when the flow state of the polymer solution changes as it flows through the partial pressure tool.

In fact, the transition from laminar to turbulent flow is not achieved simultaneously in all parts of the partial pressure tool. Usually, the streamline of the most turbulent fluid layer is the first to bend. As the intensity of turbulence increases, the fluctuation of the streamline increases and the fluid changes from fluctuation to spiral motion and eventually to vortex motion, and as the turbulence intensity continues to increase, the vortex motion accelerates until it becomes turbulent. This is consistent with the research theory of Manneville [31], and moreover similar results were obtained by Gavrilov in a study of the turbulent flow of power-law fluids in circular tubes [32]. In the partial pressure tool, the main change in the transition from laminar to turbulent flow occurs in the fluid layer with the most turbulent annular section. The maximum Reynolds number of the fluid in this layer ${\left({Re}_{r}\right)}_{m}$, which is the stability parameter Z, was used to judge the flow state. Based on the stability parameter, the velocity equation of the flow state transition is established, and is termed the transition velocity equation.

Based on the centerline of any liquid layer in an annular flow in the partial pressure tool, the Reynolds number at position r from the centerline is given by:(35)$${Re}_{r}=\frac{\rho ur}{\mu }$$
where:(36)$$\mu =k{\left(\dot{\gamma }\right)}^{n-1}=k{\left(\frac{\Delta p}{kl}\right)}^{\frac{n-1}{n}}r^{\frac{n-1}{n}}$$

Putting Equations (8) and (36) into the Reynolds number calculation Equation (35) gives: (37)$${Re}_{r}=\frac{\rho }{k}\frac{n}{n+1}{\left(\frac{\Delta p}{kl}\right)}^{\frac{2-n}{n}}\left\{{\left[\frac{1}{2}(R_{o}-R_{i})\right]}^{\frac{n+1}{n}}r^{\frac{1}{n}}-r^{\frac{n+2}{n}}\right\}$$

Solve the maximum value of ${Re}_{r}$:
(38)$$\frac{d{Re}_{r}}{dr}=0$$
(39)$$r={\left[\frac{1}{\left(n+2\right)}\right]}^{\frac{n}{1+n}}\frac{\left(R_{o}-R_{i}\right)}{2}$$
(40)$$Z={\left({Re}_{r}\right)}_{m}=\frac{\rho }{k}\frac{n}{n+1}{\left(\frac{\Delta p}{kl}\right)}^{\frac{2-n}{n}}\left\{{\left[\frac{1}{2}(R_{o}-R_{i})\right]}^{\frac{n+1}{n}}{\left[\frac{1}{\left(n+2\right)}\right]}^{\frac{1}{1+n}}{\left(\frac{R_{o}-R_{i}}{2}\right)}^{\frac{1}{n}}-{\left[\frac{1}{\left(n+2\right)}\right]}^{\frac{n+2}{1+n}}{\left(\frac{R_{o}-R_{i}}{2}\right)}^{\frac{n+2}{n}}\right\}$$

The average velocity of the polymer solution in the cross-section of the partial pressure tool:(41)$$v=\frac{Q}{\pi (R_{o}^{2}-R_{i}^{2})}$$

Putting Equation (11) into Equation (41) gives:(42)$$v=(\frac{n}{2n+1}){\left(\frac{\Delta p}{kl}\right)}^{\frac{1}{n}}{\left(\frac{R_{o}-R_{i}}{2}\right)}^{\frac{n+1}{n}}$$

Equation (42) is connected with the stability parameter Z, and then variable substitution is performed:(43)$$v={\left\{\frac{Z\frac{n+1}{n}\frac{k}{\rho }{\left(\frac{2n+1}{n}\right)}^{n-2}{\left(\frac{2}{R_{o}-R_{i}}\right)}^{\frac{\left(n-2\right)\left(n+1\right)}{n}}}{{\left(\frac{R_{o}-R_{i}}{2}\right)}^{\frac{n+2}{n}}\left[{\frac{1}{\left(n+2\right)}}^{\frac{1}{1+n}}-{\frac{1}{\left(n+2\right)}}^{\frac{2+n}{1+n}}\right]}\right\}}^{\frac{1}{2-n}}$$

Equation (43) is the transition velocity of the polymer solution in the partial pressure tool, which is related to the characteristics of the polymer solution and the structural parameters of the partial pressure tool. When the characteristics of the polymer solution and the structural parameters of the partial pressure tool are determined, the transition velocity, which is a fixed value, can be determined. When the average velocity of the polymer solution flowing through the cross-section of the partial pressure tool exceeds the transition velocity, the flow state of the polymer solution changes from a laminar to a turbulent flow state, and consequently the energy loss and the pressure drop of the polymer solution increase.

## 3. Numerical Solution of Polymer Solution Flow in the Partial Pressure Tool

The consistency coefficient and rheological index of the shear flow equation were obtained by fitting based on Figure 3, as shown in Table 4.

According to factors such as actual manufacturing difficulty and economy of partial pressure tools, Daqing oilfield provides structural parameters of several partial pressure tools. Specific structural parameters of the partial pressure tool are shown in Table 5, and the meanings of the parameters are the same as those of Figure 4.

The constitutive equation of the polymer solution, the consistency coefficient and rheological index of the shear flow equation obtained by actual fitting, and the structural parameters of the partial pressure tool, are put into the obtained flow characteristic equation, and specific numerical solutions are calculated in order to study the effect of different structural parameters on the pressure drop and viscosity of the polymer solution. Then, the best structural parameters to solve the transition velocity are selected, and the limit value of the flow transition is obtained. The calculated pressure and apparent viscosity of the polymer solution in the partial pressure tools are plotted, as shown in Figure 10 and Figure 11.

The inlet pressure of the polymer solution entering the partial pressure tool was 1.8 MPa, the injection amount was 50 m^3^/d, the front groove angles/rear groove angles of the partial pressure tool were 30°/45°, 45°/30°, and 60°/15°, and the outer cylinder radiuses of the partial pressure tool were 24, 24.5, and 25 mm. By putting the variables into the pressure drop equation, the pressure curve of the polymer solution in the partial pressure tool was obtained, as shown in Figure 10.

As can be seen from Figure 10, when the polymer solution passes through the front and rear annular flow channels, the overall pressure of the polymer solution decreases (however, the decrease is extremely small, and is therefore hard to discern in the figure). However, when the polymer solution passes through the throttling section, the pressure of the solution drops rapidly and to a larger degree, while the trend of the pressure drop changes periodically. This is due to the fact that during the flow of the solution, the polymer molecular chains are always in a deformation process of elongation and contraction, so that part of the flow energy is consumed by the deformation and recovery of the chains, which generates local energy loss and pressure reduction. This clearly shows that the partial pressure tool can reduce the pressure of the polymer solution.

The influence of different variables on the pressure is also different. For example, when the front groove angle is increased, and the rear groove angle consequently decreases, the pressure drop of the polymer solution first becomes larger and then smaller, after flowing through the whole throttling section, and the total pressure drop is the smallest for a front groove angle of 60°. This is due to the fact that, during the solution flow, a larger front groove angle and lower rear groove angle results in the polymer solution flowing through a smaller cross-section area in the contraction section of the throttling section; the more obvious the throttling effect on the solution, the larger the pressure drop of the solution in the initial stage. However, increasing the front groove angle and decreasing the rear groove angle also reduces the number of throttling areas through which the solution passes, resulting in a smaller overall pressure drop; thus, the accumulated pressure drop is at the minimum after the polymer solution passes through the throttling section.

When the radius of the outer cylinder increases, the pressure drop in each throttling section decreases, and, for the maximum radius of 25 mm, the total pressure drop after the solution completely flows through the throttling section is the smallest for any of the partial pressure tool structural parameters investigated in this study. This is due to the fact that, with increasing outer cylinder radius, the minimum cross-section area through which the polymer solution flows increases, the flow rate reduces, and the energy consumed by the solution reduces, resulting in a reduction of solution pressure loss.

From Figure 10, it can be seen that, for front groove angles/rear groove angles of the partial pressure tool of 30°/45°, 45°/30°, and 60°/15°, the pressure drop of the polymer solution reaches 1.77, 1.40, and 1.07 MPa, respectively, and for outer cylinder radiuses of 24, 24.5, and 25 mm, the pressure drop reaches 1.32, 1.10, and 0.846 MPa, respectively. In summary, when the front groove angle is 30°, the rear groove angle is 45°, and the outer cylinder radius is 24 mm, the pressure drop of the polymer solution is the largest and the throttling effect is the best.

The setting of parameters and the selection of variables are consistent with the pressure calculation. Viscosity curves of the polymer solution flowing through partial pressure tools with different structural parameters are shown in Figure 11. It can be seen that the apparent viscosity of the polymer solution before and after flowing through the partial pressure tools does not change, with changes only occurring while the solution flows through the throttling section. This change in viscosity is due to the fact that the polymer molecules are distributed in aqueous solution in the form of particles, dendritic structures, and network structures, and the molecular chains are flexible chain structures. When the flow cross-section area of the solution decreases during the flow process, the flow speed increases and the polymer molecules are elongated along the flow direction. When the polymer molecules are elongated to a certain extent and exceed the strength of the molecular chain, the shear stress acting on the molecular chain can cause changes in the shape and size of the polymer molecules, resulting in a decrease in the molecular weight of the polymer and a consequent decrease in the viscosity of the polymer solution. However, when the cross-sectional flow area decreases, the molecular chains return to their previous state due to their elasticity, resulting in an increase in solution viscosity. In the annular flow channel, the polymer molecular morphology changes similarly, so the solution viscosity does not change.

With increasing front groove angle, and consequently decreasing rear groove angle, the rate of the initial decrease in solution viscosity increases; however, the magnitude of the viscosity decrease is reduced, and the viscosity values start to recover earlier. This is due to the fact that, with increasing front groove angle and decreasing rear groove angle, the cross-section area through which the polymer solution flows in the contraction section decreases, the velocity changes sharply, the shearing effect on the polymer molecules increases, the viscosity of the solution decreases more quickly, and the polymer solution passes from the contraction section to the expansion section earlier.

With increasing outer cylinder radius, the initial viscosity of the polymer solution increases; however, the viscosity changes of the polymer solution during the flow process are basically similar for all radiuses studied. The viscosity of the polymer solution is higher for larger outer cylinder radiuses due to the fact that, with increasing outer cylinder radius, the cross-sectional area through which the polymer solution flows increases, the velocity decreases, the shearing effect on the polymer solution decreases, the degree of shearing damage to the long molecular chain of polymer molecules is consequently reduced, and the change in molecular weight change is not obvious. The change trend of polymer solution viscosity is similar for all of the studied outer cylinder radiuses due to the fact that the limited increase in radius does not lead to a variation in the shear mechanism acting on the polymer solution.

It can also be seen from Figure 11 that, for all studied parameter values, the viscosity of the polymer solution does not change after flowing through the throttling section, showing that the partial pressure tool has no influence on the viscosity of the polymer solution.

Through the analysis of pressure and viscosity curves, we have shown that the partial pressure tool has only a small influence on the viscosity of the polymer solution and can produce a large pressure drop, which meets the design purpose. Among the parameters provided, the ones which produced the largest pressure drop were a front groove angle of 30°, a rear groove angle of 45°, and an outer cylinder radius of 24 mm. Based on the transition velocity equation, the calculated transition velocity of the polymer solution is 50.29 m/s, and when the flow velocity exceeds 50.29 m/s, the flow state of the polymer solution changes from laminar flow to turbulent flow, resulting in a significant pressure drop of the polymer solution.

## 4. Conclusions

In this paper, the physical model of the partial pressure tool is established, then the flow characteristic equation and transition velocity equation of polymer solution in the partial pressure tool are deduced, and the following conclusions are obtained through specific analysis based on actual data: (1)According to the deduced rheological equation, the pressure and apparent viscosity of the polymer solution flowing through the partial pressure tools are related to the characteristics of the polymer solution and the structural parameters of the partial pressure tools;(2)When the polymer solution passes through the throttling section of the partial pressure tool, with decreasing front groove angle (and consequently increasing rear groove angle) and decreasing outer cylinder radius, the total pressure drop of the polymer solution increases. When the front groove angle and rear groove angle are 30° and 45°, respectively, the pressure drop reaches a maximum value of 1.77 MPa, and when the outer cylinder radius is 24 mm, the pressure drop reaches a maximum value of 1.32 MPa;(3)The apparent viscosity of the polymer solution is the same before and after flowing through the partial pressure tool. With increasing front groove angle (and consequently decreasing rear groove angle), the rate of the apparent viscosity decrease increases; however, the magnitude of the decrease is reduced. With increasing outer cylinder radius, the initial value of the apparent viscosity of the polymer solution increases.(4)Based on the concept of stability factor, the transition velocity equation is established, and the transition velocity of the polymer solution is calculated to be 50.29 m/s. When the velocity exceeds 50.29 m/s, the flow of the polymer solution changes from a laminar state to a turbulent state.

## Figures and Tables

**Figure 1 polymers-11-00855-f001:**
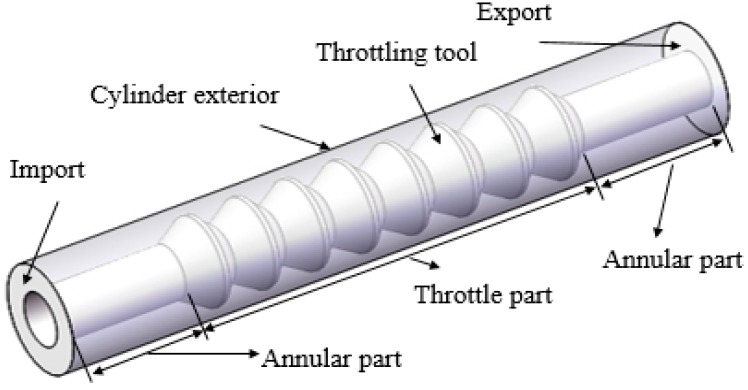
Three-dimensional representation of the partial pressure tool model.

**Figure 2 polymers-11-00855-f002:**
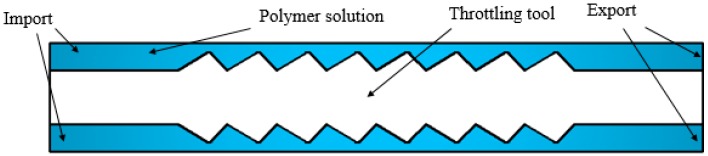
Two-dimensional representation of the partial pressure tool model.

**Figure 3 polymers-11-00855-f003:**
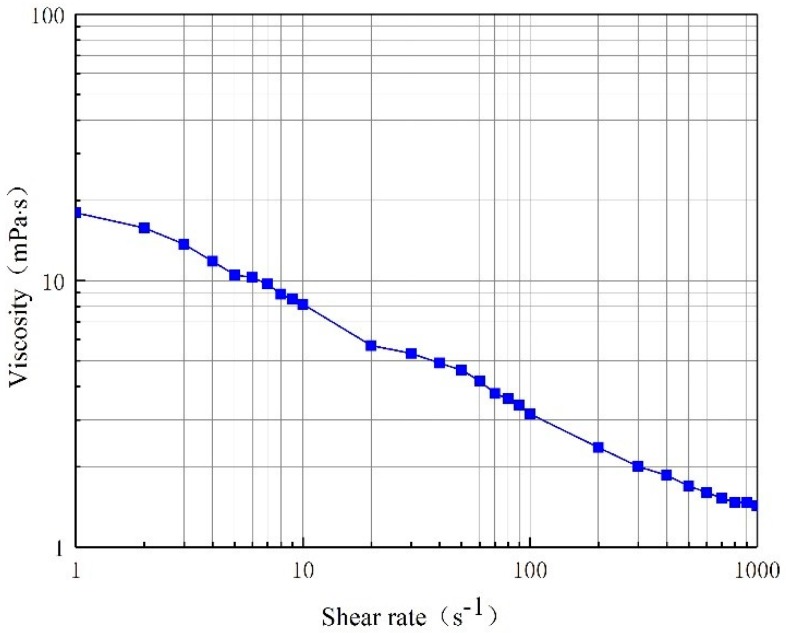
Rheological curve of the polymer solution with a polymer concentration of 1000 mg/L.

**Figure 4 polymers-11-00855-f004:**
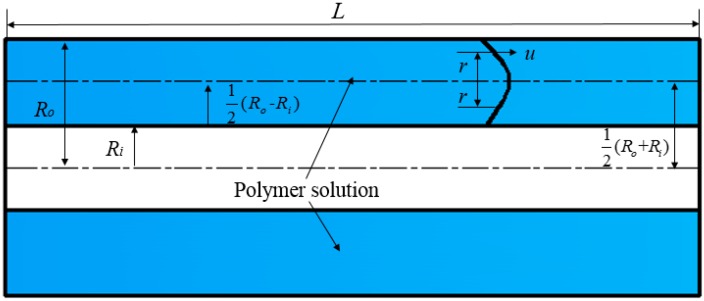
Representation of the annular flow channel with length *L*. $R_{0}$ is the radius of the outer cylinder; $R_{i}$ is the radius of the inner cylinder; r is annular flow thickness; inner diameter of annular flow = $\frac{1}{2}(R_{o}+R_{i})-r$ and outer diameter of annular flow = $\frac{1}{2}(R_{o}+R_{i})+r$.

**Figure 5 polymers-11-00855-f005:**
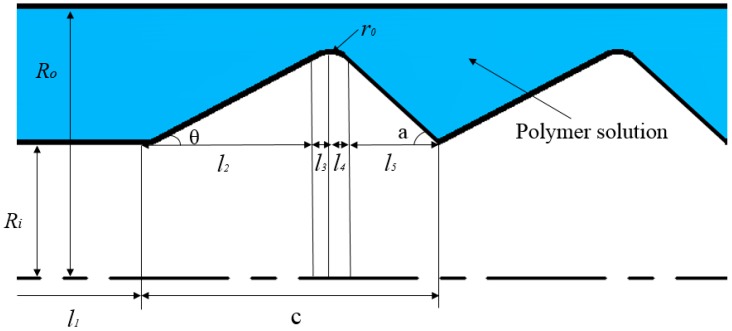
Representation of the throttling section flow channel. $R_{0}$ is the radius of the outer cylinder; $R_{i}$ is the radius of the inner cylinder; $l_{1}$ is the length of the annular flow channel; $l_{2}$ is the length of the contraction part of the single throttling section; $l_{3}$ is the length of the intermediate constricting part of the single throttling section; $l_{4}$ is the length of the intermediate expansion section of the single throttling section; $l_{5}$ is the length of the expansion section of the single throttling section; c is the total length of the single throttling section; $r_{o}$ is the radius of the intermediate arc; θ is the front groove angle of the single throttling section—it is the convex angle of the contraction section relative to the horizontal section; and α is the rear groove angle of the single throttling section—it is the convex angle of the expansion section relative to the horizontal section.

**Figure 6 polymers-11-00855-f006:**
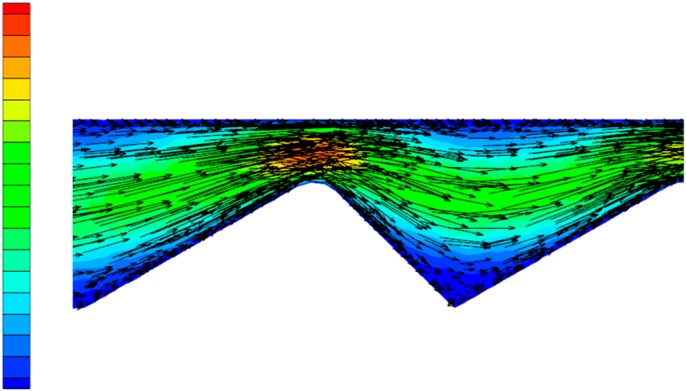
Laminar flow vectorgraph.

**Figure 7 polymers-11-00855-f007:**
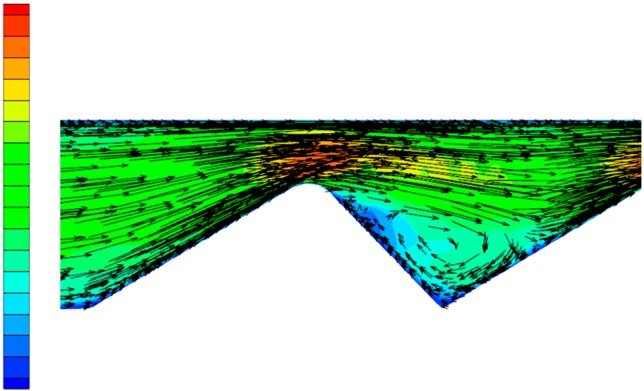
Turbulent flow vectorgraph.

**Figure 8 polymers-11-00855-f008:**
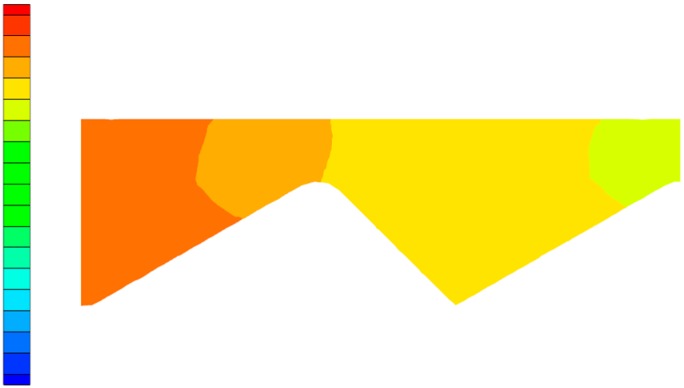
Laminar flow pressure nephogram.

**Figure 9 polymers-11-00855-f009:**
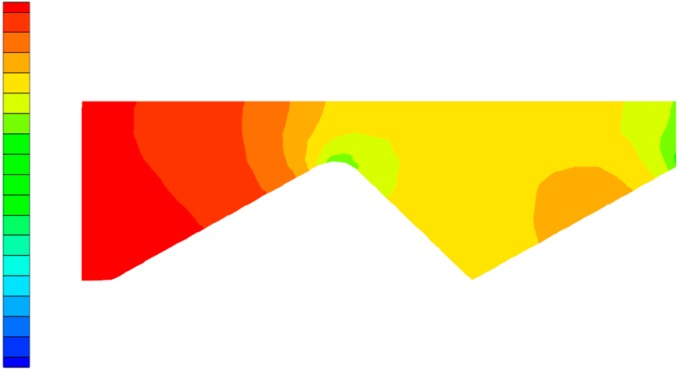
Turbulent flow pressure nephogram.

**Figure 10 polymers-11-00855-f010:**
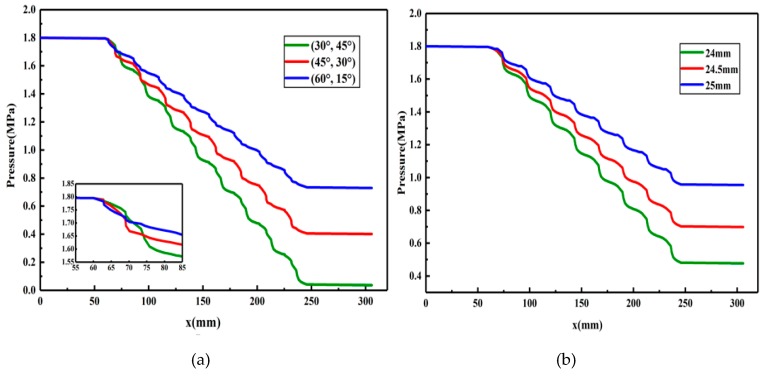
Pressure curve of the polymer solution flowing through partial pressure tools with different structural parameters. (**a**) Front groove angle/rear groove angle. (**b**) Radius of the outer cylinder.

**Figure 11 polymers-11-00855-f011:**
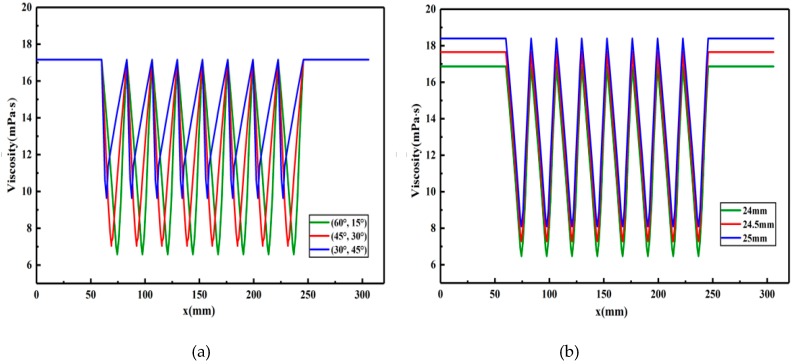
Viscosity curve of the polymer solution flowing through partial pressure tools with different structural parameters. (**a**) Front groove angle/rear groove angle. (**b**) Radius of the outer cylinder.

**Table 1 polymers-11-00855-t001:** Chemical composition of the experimental water.

Chemical Composition	Na^+^, K^+^	HCO_3_^−^	CO_3_^2−^	SO_4_^2−^	Cl^−^	Mg^2+^	Ca^2+^
**Concentration(mg/L)**	132.53	52.65	22.38	25.17	28.72	7.69	14.28

**Table 2 polymers-11-00855-t002:** Structural parameters of the annular flow channel.

Radius of the Outer Cylinder $R_{0}$ (mm)	Radius of the Inner Cylinder $R_{i}$ (mm)	Length of the Annular Flow Channel *L* (mm)
24	12	60

**Table 3 polymers-11-00855-t003:** Structural parameters of the throttling section.

Angle θ (°)	Angle α (°)	Length $l_{1}$ (mm)	Length $l_{2}$ (mm)	Length $l_{3}$ (mm)	Length $l_{4}$ (mm)	Length $l_{5}$ (mm)	Radius $r_{o}$ (mm)	Radius $R_{o}$ (mm)	Radius $R_{i}$ (mm)
30	45	60	13.392	1	1.414	7.414	2	24	12

**Table 4 polymers-11-00855-t004:** Constitutive equation of polymer solution.

Concentration (mg/L)	Consistency Coefficient k	Rheological Index *n*	Constitutive Equation
1000	20.26	0.61	$\mu =20.26{\left(\overset{\cdot }{\gamma }\right)}^{-0.39}$

**Table 5 polymers-11-00855-t005:** Structural parameters.

Angle θ (°)	Angle α (°)	Length $l_{1}$ (mm)	Length $l_{2}$ (mm)	Length $l_{3}$ (mm)	Length $l_{4}$ (mm)	Length $l_{5}$ (mm)	Radius $r_{o}$ (mm)	Radius $R_{o}$ (mm)	Radius $R_{i}$ (mm)
30	45	60	13.392	1	1.414	7.414	2	24	12
45	30	60	7.732	1.414	1	12.841	2	24	12
60	15	60	2.830	1.732	0.518	17.542	2	24	12
30	45	60	13.392	1	1.414	7.414	2	24	12
30	45	60	13.392	1	1.414	7.414	2	24.5	12
30	45	60	13.392	1	1.414	7.414	2	25	12

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
