# Peer review of "Rheological Model and Transition Velocity Equation of a Polymer Solution in a Partial Pressure Tool"

_polymers, 2019, doi:10.3390/polym11050855_

Round 1
Reviewer 1 Report
Polymers 482883.
Rheological Model and Transition Velocity Equation of Polymer Solution in Partial Pressure Tool Technical Review of Manuscript .
Bin Huang 1*, Xinyu Hu 1 , Cheng Fu 1, 2,*, Chongjiang Liu3 , Ying Wang 4 4 and Xu An
General Comments:
The article is interesting because it is novel. It deals with a topic not normally considered by researchers, but of importance in the polymer flooding rheology. The mathematical development of the rheological model presented for polymer solutions is very complete, structured and well presented. However, the authors must to improve some sections in the manuscript. In this way, some comments which must to adressed in the revision version:
1. The abstract is a bit confusing and difficult to understand, don’t you have quantitative results in your abstract? I recommend to rewrite this section. I suggest that its wording be improved to be more friendly concepts and the explanation of the main result.
2. Section 2.2 is well explained, however it is necessary to reference the statements exposed in the explanation of the rheological behavior of the polymer solutions as the shear rate increases.
3. In the mathematical development presented in section 2.4, it is necessary to clearly explain where the theta and alpha angles are measured, and whether these depend on the design of the partial pressure tool.
4. It would be wise to make a proper revision on your English spelling for upgrading the manuscript quality, e.g. you use the term “measures” while it would be more precise to use “alternatives”.
5. For the keywords it is recommended to use more accurate terms.
6. Please look at the petroleum engineering technical terminology, it is needed some corrections, e.g. in line 36 you mentioned “mining efficiency” it would be better to use “production efficiency” .
7. In the introduction you made some bibliographic review regarding on previous studies related with your research issue. The described results obtained in the mentioned studies are needed to be exposed in a more accurately way, i.e.:
“J.J. Meyers [4] injected alkaline solution into the formation for oil displacement experiment. It showed that polymer oil displacement is effective.”
How much did the displacement efficiency increase? How it rebounded in the recovery factor?
8. In the third paragraph of the introduction you mentioned “the expansion and contraction of the pores can be carried out in the pores, the oil at the edge of the pores is washed out, and the velocity gradient at the wall of the hole is high, which can make the oil film thinner.” I suggest you to revise and redefine the mechanism you mentioned for the displacement efficiency increase regarding on polymer injection, as what you mentioned is not the main mechanism, it is just an additional effect.
9. There are some other methods for increasing the displacement efficiency at reservoir conditions that are mainly used for avoiding channeling caused permeability differences between different production layers. Why do you consider that the method studied in your manuscript is of importance? Which advantages does it have over the other methods, or in which scenarios it is commonly used instead of using other methods? You need to define this in the introduction in order to show your research relevance.
10. In section 2.2. you mentioned that with an increase in shear rate further than the second asymptotic limit (second Newtonian region) it is encountered a viscoelastic response. In fact, for all the shear rate range shown in the Figure 3, it is found a viscoelastic response to shear as you have both, elastic and viscous contributions, what you really have to annotate is that the elastic contribution is higher in this shear rate range.
11. Please correct the equations formatting as there are uncompleted formulas or incorrect symbols usage.
12. It would be appropriate to stablish a methodology section, how did you carried out the rheological measurements? Which geometry and GAP were used for carrying out the tests? Did you have a temperature controller? In case you do, which is the precision of the instrument?
13. Why didn’t you used a deionized water as diluent for the polymer solution? Did you consider the polymer degradation due to the Na+ and other ions presence? The polymer degradation produced by this type of brines needs to take special attention as if you intend to scale-up the experimental results regarding the polymer rheological behavior to the flow phenomenon obtained in the partial pressure tool you will have an experimental bias as the fluid flowing behavior will be affected by rheological parameters conditioned by a previous degradation process. I suggest you to finely discriminate between these two processes to carry out a better analysis.
14. In Figure 10 you reported viscosities lower than 1 mPa.s, nonetheless, as your diluent is a type of brine, it would be expected to have viscosity values higher or at least 1 mPa.s for all the shear rate range which is approximately the sole water viscosity at the experimental conditions.
Recommendations:
I recommend publishing the paper after major revision.
Author Response
Response to Reviewer 1 Comments
First of all, thank you very much for your comments on this article. We tried our best to improve the manuscript and made some changes in the manuscript. And the replies to your comments are as follows. We appreciate for your warm work earnestly, and hope that the correction will meet with approval. Once again, thank you very much for your comments and suggestions.
Point 1: The abstract is a bit confusing and difficult to understand, don’t you have quantitative results in your abstract? I recommend to rewrite this section. I suggest that its wording be improved to be more friendly concepts and the explanation of the main result.
Response 1: Thank you very much for your valuable suggestions on the abstract part of the paper, and according to your suggestions, the language and content of the abstract part of the paper have been improved and modified to reduce the complexity of reading, to highlight the theme and experimental quantitative results in the abstract, and to carry out quantitative analysis on the pressure and apparent viscosity calculation results of polymer solution with different structural parameter partial pressure tools. The specific revision content is on the page 1 of the revised paper, lines 15-32.
We have explained for the problems of abstract as follows: “In order to solve the problem of the low production degree of oil layers caused by an excessively large permeability difference between layers during polymer flooding, we propose partial pressure injection technology using a partial pressure tool. The partial pressure tool controls the injection pressure of a polymer solution through a throttling effect to improve the oil displacement effect in high- and low-permeability reservoirs. In order to analyze the influence of the partial pressure tool on the rheological property of the polymer solution, a physical model of the tool is established, the rheological equation of the polymer solution in the partial pressure tool is established according to force balance analysis, the transition velocity equation for the polymer solution is established based on the concept of stability factor, and the influence of varying the structural parameters of the partial pressure tool on the rheological property of the polymer solution is analyzed. The results show that the pressure drop of the polymer solution increases with the decrease of the front groove angle of the partial pressure tool (from 60° to 30°), reaching a maximum of 1.77 MPa at a front groove angle of 30°. Additionally, the pressure drop of the polymer solution increases with the decrease of the outer cylinder radius (from 25 to 24 mm), reaching a maximum of 1.32 MPa at a radius of 24 mm. However, the apparent viscosity of the polymer solution before and after flowing through the partial pressure tool does not change for any of the studied parameters. These research results are of great significance to research on partial pressure injection technology and enhanced oil recovery.”
Point 2: Section 2.2 is well explained, however it is necessary to reference the statements exposed in the explanation of the rheological behavior of the polymer solutions as the shear rate increases.
Response 2: First of all, thank you very much for your affirmation of this part of the paper. For Viscoelastic section, we have consulted relevant literatures and explained that: When the shear rate increases again, the adjacent chain of the main chain deviates from the normal bond angle, thereby producing elastic restoring force and exhibiting viscoelasticity, which increases the apparent viscosity.Considering your opinion and relative to the whole paper, the polymer solution conforms to the rheological characteristics of power-law fluid through the partial pressure tool, and the power-law relationship is extremely obvious in the shear test. The equations are deduced based on the rheological equation of power-law fluid. In order to avoid the misunderstanding of the paper content caused by putting forward too many rheological models and causing disputes. Based on the original rheological curve of polymer solution, this paper only considers the rheological equations of the power-law fluid of polymer solution in the Pseudoplastic section for derivation, and does not consider other flow characteristic sections (Newtonian section, Limit Newtonian section, Viscoelastic section and Degradation section.) to make the paper more coherent and the theme and content more explicit.
The specific revision content is on the page 6 of the revised article, lines 236-242.
Point 3: In the mathematical development presented in section 2.4, it is necessary to clearly explain where the theta and alpha angles are measured, and whether these depend on the design of the partial pressure tool.
Response 3: Thank you very much for your suggestions on the model parameters. Theta and alpha angles in the partial pressure tool model are rarely introduced. They are only marked in the model structure diagram and supplemented according to the suggestions. Referring to Figure. 4, theta and alpha angles are the convex angles of the contraction and expansion sections in the throttling section of the partial pressure tool relative to the straight part of the horizontal annular flow channel. Theta and alpha angles depend on the design structure of the partial pressure tool, and the corresponding numerical values shall be determined during design. The selection of theta and alpha angles is determined through the process of optimizing structural parameters and field data in this paper. Therefore, I will make the above supplement to explain the meaning of the parameters, on the basis of the original text,
The specific revision content is on the page 8 of the revised paper, lines 299-307.
Point 4: It would be wise to make a proper revision on your English spelling for upgrading the manuscript quality, e.g. you use the term “measures” while it would be more precise to use “alternatives”.
Response 4: Thank you for your suggestions on the English vocabulary in this paper. According to your suggestions, we use "alternative" to replace "measure" effectively, modify and perfect the use of other English vocabulary in this paper, and have already made relevant rewriting and polishing on the use of the language in this paper. Specific vocabulary has been replaced in the paper.
Due to the large number of changes, other specific descriptions will not be made here.
Point 5: For the keywords it is recommended to use more accurate terms.
Response 5: Thank you very much for your suggestion on the key words in the abstract of the paper. The expression of the key words in the original manuscript is not very accurate, which has caused some misunderstanding on the content of the paper. Therefore, the choice of the key words in this paper that are more representative of the content of the paper, so that it can highlight the content of the paper. And chose partial pressure tool; rheological equation; transition velocity; pressure drop; apparent viscosity are the key words in the abstract.
Point 6: Please look at the petroleum engineering technical terminology, it is needed some corrections, e.g. in line 36 you mentioned “mining efficiency” it would be better to use “production efficiency”.
Response 6: Thank you very much for your valuable suggestions on the revision of the professional terms in this paper. According to your suggestions, the relevant technical terms of petroleum engineering in this paper have been rewritten and polished. According to your suggestions, the professional terms such as "mining efficiency", "disturbance" and others have been revised more professionally. The specific terms have been revised in the original text.
Due to the large number of changes, other specific descriptions will not be made here.
Point 7: In the introduction you made some bibliographic review regarding on previous studies related with your research issue. The described results obtained in the mentioned studies are needed to be exposed in a more accurately way, i.e.:
“J.J. Meyers [4] injected alkaline solution into the formation for oil displacement experiment. It showed that polymer oil displacement is effective.”
How much did the displacement efficiency increase? How it rebounded in the recovery factor?.
Response 7: Thank you very much for your valuable suggestions on the literature review part of the introduction in the paper. I am sorry that the original manuscript literature review part did not have a very accurate analysis and summary. According to your suggestions and revision methods, the literature review in the introduction was revised to improve the content.
We have explained for the problems of abstract as follows: “Meyers [4] carried out polymer flooding experiments in two newly developed wells by means of chemically enhanced oil recovery in Kottrabathe oilfield. In the subsequent process, specific investigations were carried out on the amount of exploitation and degree of formation utilization in the two newly developed wells. The results show that the oil recovery efficiency of polymer flooding in the total affected area was 60.6% and the oil recovery efficiency of water flooding in the total affected area was 39.9%, i.e., the effect of polymer flooding was obviously higher than that of water flooding. Clark [5] conducted polymer flooding research in the West Kiehl oilfield, and conducted oil displacement tests by means of mobility control, profile modification, and the injection of combined mobility control profile. The experiments showed that polymer flooding successfully displaced residual oil that could not be displaced during water flooding, and the oil recovery ratio increased from 11% to 40%, i.e., the recovery ratio of crude oil was greatly increased. Denney [6] concluded from experiments that, when the pressure gradient and interfacial tension of oil layers remain unchanged, polymer flooding can achieve a production efficiency that is 23.87% higher than that of water flooding under the same conditions. An oil displacement experiment was carried out by injecting high-concentration polymer solution into heterogeneous sediment cores. The results show that the higher the polymer concentration and viscosity of the displacement fluid, the higher the production efficiency. Using simulation and core flooding for enhanced oil recovery screening, Wassmuth et al. [7] proposed that, in heavy oil reservoirs, polymer flooding is the preferred enhanced oil recovery technology after primary oil recovery and water flooding. Additionally, when the water cut reaches 95% after water injection, polymer flooding can improve the oil recovery ratio by 20%, and the most practical enhanced oil recovery technology for heavy oil reservoirs is a combination of polymer flooding and horizontal wells.
Some scholars have concluded that polymer flooding can only expand the volume of the flood, and cannot improve the efficiency of oil displacement, so that the oil recovery value is lower [8]. The reason for this conclusion was that they considered the polymer solution as a Newtonian fluid [9–10]. However, in fact, polymer solutions are non-Newtonian fluids. Enhanced oil recovery is mainly allowed by mechanisms which improve increase the viscosity of, and reduce the permeability of, the water phase, which reduces the fluidity of the water phase and the fluidity ratio of water flooding, and thus improves the sweep efficiency of the flooding and achieves the purpose of enhanced oil recovery. Additionally, due to its rheological properties, the polymer exerts a stretching effect on the oil film or oil droplets in the flow process, which increases the carrying force and improves the micro-oil washing efficiency of the polymer solution. Therefore, it is of great significance to study the rheology of polymer solutions for the polymer flooding injection process.
Yin et al. [11] simulated the flow characteristics of polymer solutions at both ends of the pores in a reservoir by means of numerical simulation, drew contour lines of polymer solution velocity and flow function, and quantitatively calculated the micro-scale scanning efficiency of the solutions. The simulation results show that the sweep area and displacement efficiency increase with the increase of the viscoelasticity of the polymer solution. Compared with Newtonian fluids, the viscoelasticity of polymer solutions can improve the displacement efficiency of pores. By establishing rheological equation s, Lee [12] elaborated that the rheological property of a polymer solution for enhanced oil recovery depends on the molecular properties, concentration, salinity, shear rate, and temperature of the polymer. Rheological measurements were carried out using commonly used polymers, and rheological model parameters related to these variables were determined. The results show that with increasing temperature, salinity, and shear rate, the concentration and viscosity of the polymers decreased. At the same time, the oscillatory rheological characteristics of the polymers were measured in order to determine the flow behavior of the polymers in a reservoir rock. Silva [13] studied the rheological behavior of different polymer solutions in porous media through laboratory tests of fluid flow in porous media and analysis of fluid rheology and the hydrodynamic diameters of polymer molecules. Due to the increasing hydrodynamic volume of the polymer during the shearing and dilution of the polymer solution, the adsorption of the polymer solution was increased, indicating that polymer flooding can improve oil recovery in oilfields.”
Due to the large number of changes, other specific descriptions will be not made here.
The specific revision content is on the page 2 of the revised paper, lines 55-157.
Point 8: In the third paragraph of the introduction you mentioned “the expansion and contraction of the pores can be carried out in the pores, the oil at the edge of the pores is washed out, and the velocity gradient at the wall of the hole is high, which can make the oil film thinner.” I suggest you to revise and redefine the mechanism you mentioned for the displacement efficiency increase regarding on polymer injection, as what you mentioned is not the main mechanism, it is just an additional effect.
Response 8: Thank you very much for your comments on the oil displacement mechanism of polymer solution in the introduction part of the paper. The "sponge effect" described in the original manuscript is an additional oil displacement effect of polymer oil displacement, and cannot generally summarize the mechanism of polymer oil displacement. According to your suggestions, the mechanism of polymer oil displacement efficiency is supplemented, and relevant data and documents are reviewed and modified, which are more in line with the contents of this article.
The specific contents are as follows: “Enhanced oil recovery is mainly allowed by mechanisms which improve increase the viscosity of, and reduce the permeability of, the water phase, which reduces the fluidity of the water phase and the fluidity ratio of water flooding, and thus improves the sweep efficiency of the flooding and achieves the purpose of enhanced oil recovery. Additionally, due to its rheological properties, the polymer exerts a stretching effect on the oil film or oil droplets in the flow process, which increases the carrying force and improves the micro-oil washing efficiency of the polymer solution. Therefore, it is of great significance to study the rheology of polymer solutions for the polymer flooding injection process.”
The specific revision content is on the page 2 of the revised paper, lines 77-87.
Point 9: There are some other methods for increasing the displacement efficiency at reservoir conditions that are mainly used for avoiding channeling caused permeability differences between different production layers. Why do you consider that the method studied in your manuscript is of importance? Which advantages does it have over the other methods, or in which scenarios it is commonly used instead of using other methods? You need to define this in the introduction in order to show your research relevance
Response 9: Thank you very much for your valuable suggestions on the importance and applicability of the partial pressure tool in the introduction. The introduction of the partial pressure tool in the original manuscript is introduced unclearly and does not reflect the importance and applicability. According to your suggestions and requirements, the introduction of the partial pressure tool is supplemented, emphasizing the importance of the partial pressure tool and its adaptability under specific conditions, and necessary explanations are made in the introduction.
The specific changes are as follows: “With the development of oilfields, many alternatives have been taken to improve their production efficiency. The process of polymer flooding in oilfields can improve the injection profile to a certain extent. However, due to the influence of various factors, such as reservoir heterogeneity and well-pattern perfection, interlayer contradiction has not been fundamentally solved. Under the general injection mode, high-molecular-weight polymer enters more into high-permeability reservoirs and less into thin and poor reservoirs, which greatly reduces the permeability and production degree of thin and poor reservoirs. However, the direct injection of low-molecular-weight polymer solution can increase the degree of control of polymer flooding, although the overall flooding effect will be affected to some extent as the flooding effect of the high-permeability layer decreases. In order to improve the overall development effect of polymer flooding and improve interlayer contradiction, Daqing Oilfield has proposed a partial pressure polymer injection process. The main technical idea is to use a partial pressure tool to reduce the injection pressure of polymer solution while ensuring the oil displacement effect of high-molecular-weight polymer in the high-permeability layer, so as to control the injection amount in the high-permeability layer, make the polymer solution enter the low-permeability layer to produce oil displacement, and improve the overall development effect.”
“Although partial pressure injection technology and matching tools are proposed, which can adjust the injection pressure of polymer solution and improve the injection relationship between oil layers, these methods, while adjusting the pressure, cause a large reduction in viscosity when polymer solution is injected into a high-permeability oil layer , due to shearing action. When a high pressure drop occurs, the viscosity reduction can be as high as 40%, which seriously affects the oil displacement effect of the polymer in high- and low-permeability oil layers, preventing a maximum oil recovery rate. Therefore, it is necessary to establish a partial pressure tool that can generate a large drop in the pressure of the polymer solution while minimizing the viscosity reduction.”
“The research shows, the technologies related to polymer flooding are very mature; however, there are few researches related to improving interlayer contradiction, controlling the injection amount of polymer solution in high- and low-permeability oil layers, and further improving the oil displacement effect of polymer flooding. Only the Daqing Oilfield has proposed the concept of partial pressure polymer solution injection technology to solve this problem. Existing theories and tools can be used to adjust the pressure of polymer solutions, however they produce large reductions in the viscosity of the solutions and affect their oil displacement effect. Additionally, the rheological properties of polymer solutions in partial pressure tools have not yet been studied. The establishment of mathematical models for the rheological properties of polymer solutions in partial pressure tools mathematical model is relatively single sample, lacks theoretical basis, and does not consider the problem of fluid state transition. Therefore, the establishment of rheological models of partial pressure tools that produce large pressure drops and low viscosity reductions of polymer solutions is extremely important for enhanced oil recovery in oilfields.”
The specific revision content is on the page 2 of the revised paper, lines 39-54, page 3 of the revised paper, lines 106-121, and page 4 of the revised paper, lines 158-170.
Point 10: In section 2.2. you mentioned that with an increase in shear rate further than the second asymptotic limit (second Newtonian region) it is encountered a viscoelastic response. In fact, for all the shear rate range shown in the Figure 3, it is found a viscoelastic response to shear as you have both, elastic and viscous contributions, what you really have to annotate is that the elastic contribution is higher in this shear rate range.
Response 10: Thank you very much for your suggestions and explanations on the modification of the rheological curve of polymer solution. Consider your suggestions " for all the shear rate range shown in the Figure 3, it is found a viscoelastic response to shear as you have both, elastic and viscous contributions, what you really have to annotate is that the elastic contribution is higher in this shear rate range." and make the following analysis of this part of the content relative to the content of the whole paper. As the polymer solution shows different rheological properties with the increase of shear rate under the action of shear, considering your reference opinions and researches in related fields, the main parameters affecting viscoelasticity include storage modulus and loss modulus, and it is found that with the increase of shear rate, both storage modulus and loss modulus increase, but the increase degree of storage modulus is greater than loss modulus. When it increases to a certain extent, storage modulus will be greater than loss modulus, making the influence of elasticity on polymer solution greater than that of viscosity on polymer solution, so the contribution of elasticity is higher. However, for this paper, the polymer solution conforms to the rheological characteristics of power-law fluid through the partial pressure tool, and the power-law relationship is extremely obvious in the shear test. Moreover, the equation is deduced based on the rheological equation of power-law fluid. In order to avoid the misunderstanding of the article content caused by too many rheological models and cause disputes, based on the original rheological curve of polymer solution, this paper only considers the rheological equation of the power-law fluid of polymer solution in the Pseudoplastic section for derivation, and does not consider other flow characteristic sections (Newtonian Section, Limit Newtonian section, Viscoelasticity section and Degradation section.) to make the article more coherent, and the theme and content more explicit.
Point 11: Please correct the equations formatting as there are uncompleted formulas or incorrect symbols usage.
Response 11: Thank you very much for your suggestions on the format and symbol use of the formula in the paper. According to your suggestions, the formula and symbol have been modified and supplemented, and the order of magnitude of variables have been supplemented, so that the expression of physical quantities and equations is more accurate and meets the requirements of the article. The equations in this paper are edited by Mathtype software, and some equations may not be completely displayed due to the format problem of the transmission file, so PDF format was added when uploading the manuscript.
Point 12: It would be appropriate to stablish a methodology section, how did you carried out the rheological measurements? Which geometry and GAP were used for carrying out the tests? Did you have a temperature controller? In case you do, which is the precision of the instrument?
Response 12: Thank you for your suggestions on the experimental part of the paper. The description of the experiment in the original manuscript and the setting of the chapters need to be improved. In this paper, only the shear test of polymer solution is carried out, and the solution of pressure and apparent viscosity is calculated through the deduced rheological equation. So, according to your suggestions, the experimental part is modified and the contents of the chapters are adjusted. The experimental part is put into the methodology section. The specific contents of the experimental operation are modified as follows:
Experimental chemicals:
The polymer used to prepare the polymer solution in this experiment was partially hydrolyzed polyacrylamide (HPAM) with a relative molecular mass of 1600 × 104, a degree of hydrolysis of 26%, and a mass fraction of 90.17%. The HPAM was provided by the Daqing Oilfield Production Technology Research Institute. The experimental water used for preparing the polymer solution was provided by the No. 1 Oil Production Plant of the Daqing Oilfield; a filter membrane was required before use.
Experimental instruments:
YP-B2003 electronic balance (Huaguang instrument factory, Liaoyang, China); EURO-ST D S25 agitator (IKA, Staufen, Germany); RS-150 rheometer (IKA, Staufen, Germany).
Experimental method:
1. The YP-B2003 electronic balance was used to measure dry powder of the HPAM and experimental water in proportion to prepare a polymer solution with a concentration of 1000 mg/L;
2. The polymer mother liquor was dissolved and agitated using the EURO-ST D S25 electronic agitator at 250 r/min for 2.5 hours and then left to stand for 2 hours to ensure full dispersion of the solution molecules and a uniform system;
3. The thermostatic circulation system was started in the RS-150 rheometer, heated to 45 °C, and put in homeostasis for 15 min. The prepared solution was put into a preheated measuring outer cylinder, and the temperature was kept constant for 20 min so that the temperature of each point of the sample could reach the testing temperature;
4. The shear rate was set from 1 to 1000 s-1, the rheometer was started, and the rheometer’s viscosity option was selected for testing. When the indication viscosity value was basically stable, recording was started, and then recording was performed every 5 minutes. Four viscosity values were recorded continuously, and if the deviations between the first value and the three other values did not exceed 5%, the system was considered to have reached dynamic equilibrium.
The most important instrument in the experiment is RS-150 rheometer, its main technical indexes are:
Frequency range: 1×10-5-100.0HZ
Rotation velocity: 6.283×10-5-628.3rad/s
Torque range: 0.02-2×10-5
Temperature range: -50-500℃
The specific revision content is on the page 4 of the revised paper, lines 186-210.
Point 13: Why didn’t you used a deionized water as diluent for the polymer solution? Did you consider the polymer degradation due to the Na+ and other ions presence? The polymer degradation produced by this type of brines needs to take special attention as if you intend to scale-up the experimental results regarding the polymer rheological behavior to the flow phenomenon obtained in the partial pressure tool you will have an experimental bias as the fluid flowing behavior will be affected by rheological parameters conditioned by a previous degradation process. I suggest you to finely discriminate between these two processes to carry out a better analysis.
Response 13: Thank you very much for your suggestions on the use of brine in the experimental part. According to your suggestions, a literature survey was conducted before the experiment was conducted. From the literature, we know that salinity has a very important effect on the apparent viscosity, because the cations in it can shield the negative charge of carboxyl groups on the polymer chain, so that the molecular chain shrinks with the decrease of electrostatic repulsion, resulting in the decrease of the viscosity of the polymer solution.
At the beginning of this paper, deionized water was selected to carry out experiments to study the rheological properties of polymer solution in partial pressure tools. However, the formation water in Daqing Oilfield is not deionized water in its true state, but saline water. Therefore, in order to meet the actual production requirements of Daqing Oilfield and more truly simulate the oil displacement effect of polymer solution in the formation, we chose saline water as diluent to prepare the polymer solution.
Although cations will affect the viscosity of polymer solution, this paper only verifies the influence of pressure and apparent viscosity before and after the prepared polymer solution flows through the partial pressure tool, so it can be considered that the polymer solution is affected by cations before and after flowing through the partial pressure tool, and this condition can be taken as the default condition, and the purpose of this study is to verify the pressure drop that the partial pressure tool can generate. And whether the viscosity value of the solution has obvious change before and after flowing through the partial pressure tool. Under the condition of ensuring that the polymer solution is acted by the same ions, the acting effect of ions on the polymer solution can be ignored, and only the overall viscosity change of the polymer solution in the partial pressure tool can be concerned.
However, for the preparation of polymer solution with deionized water, although the values of pressure and viscosity will change when the polymer solution flows through the partial pressure tool compared with the polymer solution prepared with brine, the change trend of pressure and apparent viscosity are the same, the pressure drop generated under the action of the partial pressure tool is relatively small, and the apparent viscosity before and after flowing through the partial pressure tool has not changed, which is consistent with the conclusion of the effect of the partial pressure tool on the rheological property of polymer solution obtained from the experiment with brine.
Therefore, based on the above analysis, the experiment with polymer solution prepared with brine can conform to the real experimental environment, and can simulate more real experimental results under the same action conditions.
Point 14: In Figure 10 you reported viscosities lower than 1 mPa.s, nonetheless, as your diluent is a type of brine, it would be expected to have viscosity values higher or at least 1 mPa.s for all the shear rate range which is approximately the sole water viscosity at the experimental conditions.
Response 14: Thank you very much for pointing out this problem. After measuring the viscosity of polymer solution with RS-150 Rheometer, in the shear range of 1-1000s-1, the corresponding relationship between shear rate and viscosity was obtained. However, when the data was counted and converted with logarithmic coordinates, an error occurred when inputting the data, which led to an error in the rheological curve drawn, making the minimum viscosity value after shearing less than 1 MPa s. I am very sorry for this. According to your suggestion, the rheological curve according to the actual measurement results is redrew, the transformation form of logarithms is paid attention to, then a new curve equation is fitted to obtain the corrected consistency coefficient and rheological index of polymer solution rheological curve. The newly obtained consistency coefficient, rheological index and structural parameters are substituted into the deduced rheological equation, the apparent viscosity and pressure are re-solved to obtain a new numerical solution, the curves of pressure and apparent viscosity are re-drew and analyzed, and then the transition velocity with the optimized data is re-solved.
The specific revision content is on the page 13 of the revised paper, lines 436-443, and the page 14 of the revised paper, lines 447-542.
Reviewer 2 Report
The present manuscript describes the flow of a non-Newtonian liquid inside a pipe with a special profile that would be used in enhanced oil recovery, which the authors call a "partial pressure injection tool".
I think that this manuscript requires major revisions before publication, possibly in another journal.
Overall, the language has to be improved, because it obstructs the understanding.
The adequation to the journal Polymers is poor in my opinion, although a very similar article was published by the same authors in Polymers 2 months ago [Polymers 2019,11, 319]. Indeed, this manuscript mostly deals with flow of a non-Newtonian fluid in a particular geometry. The relation with polymer science is not obvious. A journal about rheology or oil recovery would be better suited.
The introduction explains the context of oil recovery and polymer flooding in many (often unnecessary) details, but does not explain clearly the point of using such a shaped pipe, nor the reason for such a strange name (partial pressure injection tool). It is not clear if the objective is to have a given pressure drop, a high flow without turbulence, a low shear rate, less energy dissipation or whatever...
The experimental part should be more detailed: what instrument was used for what measure under which conditions? The authors should consider to put the experimental part first.
The tables should specify whether the results are obtained experimentally or by computations. Tables 2 & 3 should specify the units.
Figures 7 (likely 6 also) are not obtained by the present model. Is this just an illustration or have finite element simulations been run?
Indeed, this type of computation seems particularly suited to finite element computations. This would yield more accurate results, insensitive to the rough approximations that are tacitly made in the flow model (flow symmetric in r, results for parallel flow used in non-parallel geometry, turbulence criterion for a parallel flow), and the impact of which is not discussed.
The orders of magnitude of the dimensions and quantities (Ri, Ro, L, C, Delta p, flow) should be given from the beginning.
The details about the rheological behavior of the solution in regimes other than "pseudoplastic" are not justified. The power law dependence of viscosity with shear rate is clear in the experiment, so the authors need (should) not propose debatable models for the other regimes.
The turbulence of the flow should be distinguished from the presence of a vortex.
Author Response
Response to Reviewer 2 Comments
First of all, thank you very much for your comments on this article. We tried our best to improve the manuscript and made some changes in the manuscript. And the replies to your comments are as follows. We appreciate for your warm work earnestly, and hope that the correction will meet with approval. Once again, thank you very much for your comments and suggestions.
Point 1: Indeed, this manuscript mostly deals with flow of a non-Newtonian fluid in a particular geometry. The relation with polymer science is not obvious. A journal about rheology or oil recovery would be better suited.
Response 1: First of all, thank you very much for your suggestions of our paper. The highly vertical heterogeneities of reservoirs in China lead to increasing difficulty in injection development. Polymer flooding has been used in Daqing oilfield to improve the contradiction between layers in the process of oilfield development. Many studies show that the recovery efficiency of oil field can be improved significantly by improving the rheology property of the polymer solution. It is of great significance to the study of polymer flooding. Therefore, the polymer plays an irreplaceable role in our paper. In addition, based on polymer solution, this paper studies the rheology of polymer solution in partial pressure tools, which is of great significance to study the rheology of polymer solution and the problem of enhancing oil recovery in oilfield production.
Therefore, we sincerely hope that you can reconsider our paper.
Point 2: The introduction explains the context of oil recovery and polymer flooding in many (often unnecessary) details, but does not explain clearly the point of using such a shaped pipe, nor the reason for such a strange name (partial pressure injection tool). It is not clear if the objective is to have a given pressure drop, a high flow without turbulence, a low shear rate, less energy dissipation or whatever...
Response 2: Thank you very much for your valuable suggestions on the introduction of the paper. In the introduction part of the paper, the background of polymer flooding is first introduced in detail. The detailed introduction in this part is to draw out the influence of partial pressure tools on polymer flooding from the perspective of polymer flooding. As the concept of partial pressure injection technology has not been widely involved in oil displacement technology, the use and research of partial pressure tools are also in the state of theoretical research. Therefore, the research background of polymer flooding is introduced in detail, and the problems existing in the process of polymer flooding are expounded, which provides a theoretical basis for studying the application of partial pressure tools in polymer flooding and better studies the mechanism of partial pressure tools on polymer solution to improve the crude oil recovery rate of polymer flooding. However, the introduction of the partial pressure tools in the original manuscript is vague and does not emphasize the important role of the tools. According to your suggestions and requirements, the introduction of the relevance of the partial pressure tools is supplemented, emphasizing the importance of the partial pressure tools and its adaptability under specific conditions. Necessary explanations are made in the introduction.
The specific changes are as follows: “With the development of oilfields, many alternatives have been taken to improve their production efficiency. The process of polymer flooding in oilfields can improve the injection profile to a certain extent. However, due to the influence of various factors, such as reservoir heterogeneity and well-pattern perfection, interlayer contradiction has not been fundamentally solved. Under the general injection mode, high-molecular-weight polymer enters more into high-permeability reservoirs and less into thin and poor reservoirs, which greatly reduces the permeability and production degree of thin poor reservoirs. However, the direct injection of low-molecular-weight polymer solution can increase the degree of control of polymer flooding, although the overall flooding effect will be affected to some extent as the flooding effect of the high-permeability layer decreases. In order to improve the overall development effect of polymer flooding and improve interlayer contradiction, Daqing Oilfield has proposed a partial pressure polymer injection process. The main technical idea is to use a partial pressure tool to reduce the injection pressure of polymer solution while ensuring the oil displacement effect of high-molecular-weight polymer in the high-permeability layer, so as to control the injection amount in the high-permeability layer, make the polymer solution enter the low-permeability layer to produce oil displacement, and improve the overall development effect.”
“Although partial pressure injection technology and matching tools are proposed, which can adjust the injection pressure of polymer solution and improve the injection relationship between oil layers, these methods, while adjusting the pressure, cause a large reduction in viscosity when polymer solution is injected into a high-permeability oil layer , due to shearing action. When a high pressure drop occurs, the viscosity reduction can be as high as 40%, which seriously affects the oil displacement effect of the polymer in high- and low-permeability oil layers, preventing a maximum oil recovery rate. Therefore, it is necessary to establish a partial pressure tool that can generate a large drop in the pressure of the polymer solution while minimizing the viscosity reduction.”
“Research shows, the technologies related to polymer flooding are very mature; however, there are few researches related to improving interlayer contradiction, controlling the injection amount of polymer solution in high- and low-permeability oil layers, and further improving the oil displacement effect of polymer flooding. Only the Daqing Oilfield has proposed the concept of partial pressure polymer solution injection technology to solve this problem. Existing theories and tools can be used to adjust the pressure of polymer solutions, however they produce large reductions in the viscosity of the solutions and affect their oil displacement effect. Additionally, the rheological properties of polymer solutions in partial pressure tools have not yet been studied. The establishment of mathematical models for the rheological properties of polymer solutions in partial pressure tools mathematical model is relatively single sample, lacks theoretical basis, and does not consider the problem of fluid state transition. Therefore, the establishment of rheological models of partial pressure tools that produce large pressure drops and low viscosity reductions of polymer solutions is extremely important for enhanced oil recovery in oilfields.”
The specific revision content is on the page 1 of the revised paper, lines 39-54, page 3 of the revised paper, lines 122-130, and page 4 of the revised paper, lines 158-170.
Point 3: The experimental part should be more detailed: what instrument was used for what measure under which conditions? The authors should consider to put the experimental part first.
Response 3: Thank you for your valuable suggestions on the experimental part of the paper. The description of the experiment in the original manuscript and the setting of the chapters need to be improved. According to your suggestions, the experimental part is improved and modified, and the contents of the chapters are adjusted. The experimental part is put into the methodology section. The specific and perfect contents of the experimental operation are as follows:
Experimental chemicals:
The polymer used to prepare the polymer solution in this experiment was partially hydrolyzed polyacrylamide (HPAM) with a relative molecular mass of 1600 × 104, a degree of hydrolysis of 26%, and a mass fraction of 90.17%. The HPAM was provided by the Daqing Oilfield Production Technology Research Institute. The experimental water used for preparing the polymer solution was provided by the No. 1 Oil Production Plant of the Daqing Oilfield; a filter membrane was required before use. The composition of the experimental water is shown in Table 1.
Experimental instruments:
YP-B2003 electronic balance (Huaguang instrument factory, Liaoyang, China); EURO-ST D S25 agitator (IKA, Staufen, Germany); RS-150 rheometer (IKA, Staufen, Germany).
Experimental method:
1. The YP-B2003 electronic balance was used to measure dry powder of the HPAM and experimental water in proportion to prepare a polymer solution with a concentration of 1000 mg/L;
2. The polymer mother liquor was dissolved and agitated using the EURO-ST D S25 electronic agitator at 250 r/min for 2.5 hours and then left to stand for 2 hours to ensure full dispersion of the solution molecules and a uniform system;
3. The thermostatic circulation system was started in the RS-150 rheometer, heated to 45 °C, and put in homeostasis for 15 min. The prepared solution was put into a preheated measuring outer cylinder, and the temperature was kept constant for 20 min so that the temperature of each point of the sample could reach the testing temperature;
4. The shear rate was set from 1 to 1000 s-1, the rheometer was started, and the rheometer’s viscosity option was selected for testing. When the indication viscosity value was basically stable, recording was started, and then recording was performed every 5 minutes. Four viscosity values were recorded continuously, and if the deviations between the first value and the three other values did not exceed 5%, the system was considered to have reached dynamic equilibrium.
The specific revision content is on the page 4 of the revised paper, lines 187-211.
Point 4: The tables should specify whether the results are obtained experimentally or by computations. Tables 2 & 3 should specify the units.
Response 4: Thank you for your suggestions on the table section of the paper, and make changes according to your suggestions. Table 1 shows the specific composition and content of experimental water used to prepare polymer solution, which is provided by No 1 Oil Production Plant of Daqing Oilfield, unit is mg/L. Table 2 is the rheological equation of polymer solution, which is calculated by fitting the rheological curve of polymer solution obtained from shear rate and viscosity of polymer solution measured by RS-150 Rheometer. The data in the table include concentration, unit is mg/L, k and n are consistency coefficient and rheological index respectively, both are dimensionless numbers, and the fitted rheological curve equation is the calculation result. Table 3 shows the structural parameters of the partial pressure tools, which is used to study the flow characteristics of polymer solution in the partial pressure tools with different structural parameters. The data in the table are selected according to the actual production data in the field polymer flooding process and the structural parameters of the matching tools of the partial pressure injection technology.
The specific revision content is on the page 5 of the revised paper, line 212, the page 13 of the revised paper, line 443, and the page 14 of the revised paper, line 446.
Point 5: Figures 7 (likely 6 also) are not obtained by the present model. Is this just an illustration or have finite element simulations been run?
Response 5: Thank you for your suggestion on the vectorgraph and pressure nephogram for simulating laminar flow and turbulent flow, which was obtained by finite element simulation. In order to more clearly observe the flow state of polymer solution from laminar flow to turbulent flow in the partial pressure tool and the influence of flow state change on pressure, Fluent software is used for numerical simulation. During the simulation process, the model structure does not change, and the vector diagram and pressure nephogram of polymer solution in laminar flow and turbulent flow state are obtained. As the critical velocity from laminar flow to turbulent flow cannot be accurately obtained during the simulation process, Therefore, the flow velocity is gradually increased and set. According to the vector diagram and nephogram drawn by simulation, the flow state from laminar flow to turbulent flow and the change of pressure can be observed more intuitively. The rheological equation formula deduced cannot intuitively express these phenomena. Therefore, the use of Fluent software is not introduced too much in this part, but only the finite element simulation is carried out to illustrate the phenomenon of flow state transition. On this basis, the critical velocity of flow state transition is deduced and solved by mathematical derivation.
The specific revision content is on the page 11 of the revised paper, lines 356-368.
Point 6: Indeed, this type of computation seems particularly suited to finite element computations. This would yield more accurate results, insensitive to the rough approximations that are tacitly made in the flow model (flow symmetric in r, results for parallel flow used in non-parallel geometry, turbulence criterion for a parallel flow), and the impact of which is not discussed.
Response 6: Thank you for your suggestions to use finite element analysis and calculation for the solution part of the paper. Combined with your suggestions, this paper mainly describes the rheological study of polymer solution in partial pressure tools, while the research and application of partial pressure tools is rare. Currently, only Daqing Oilfield is involved in the process of polymer flooding, but it is also in the research and development and experimental stages. The prospect of large-scale application and production is still unknown. The common methods to study the influence of partial pressure tools on polymer solution mainly include finite element analysis and calculation and formula derivation of rheological model.
For this model, finite element analysis and calculation can produce more accurate results, and is not affected by the flow state transition of polymer solution in the partial pressure tool. However, it requires higher accuracy of physical model and higher requirements for division of model grids. When the number of grids in the model is large enough, the obtained results are more accurate, but the simulation time is longer, and the finite element calculation results cannot be obtained within a limited time. When the partial pressure tools with different structures are to be simulated, the time spent will be multiplied, and the requirements and methods for finite element calculation will be more stringent, so the simulated results cannot be obtained within a limited time, and the requirements of oil fields cannot be met.
Compared with the finite element calculation, the formula derivation of the rheological model is lower in accuracy and requires higher requirements for the rheological state transformation. However, for the simulation and analysis of the influence trend of the partial pressure tool on the rheological property of polymer solution, the calculation amount is less, and the rheological change can be analyzed within a limited time. For the partial pressure tools with different structures, it is easier to adjust the structural parameters and can adapt to different situations. In actual production of oil fields, in order to ensure the simulation efficiency and the requirements of rapid production simulation of oil fields, it is reasonable to save the simulation calculation time and select the formula derivation of the rheological model for verification.
Point 7: The orders of magnitude of the dimensions and quantities (Ri, Ro, L, C, Delta p, flow) should be given from the beginning.
Response 7: Thank you very much for your suggestions on the size and magnitude of the tool parameters. According to your suggestions, the magnitude of each parameter is supplemented in the paper, which makes the understanding of physical quantities clearer. In the derivation of rheological equation and transition flow rate of the partial pressure tools for polymer solution, the size parameter of the partial pressure tool is a generalized concept, which can meet the accuracy and universality of rheological model and transition flow rate with different structural parameters. Therefore, it is not necessary to substitute specific size parameters for calculation in this process, but when carrying out the influence of different structural parameters on pressure drop and apparent viscosity, it is necessary to bring in specific size values of partial pressure tools to more intuitively analyze the changes of pressure and apparent viscosity, and at the same time optimize the structural parameters to provide theoretical basis for actual production process. Therefore, the size values of partial pressure tools are given before substituting into numerical calculation, making the structure of the whole paper more coherent. And table 3 lists the different structural parameters in the partial pressure tool, and the specific meaning of the parameters has been specifically explained in Figure. 4, so the meaning of the parameters in table 3 need not be explained.
And before the polymer solution enters the throttling section of the partial pressure tool, the solution flows in the annular flow channel, as shown in Figure 43. It is assumed that the flow of the polymer solution always conforms to the flow characteristics of thea power-law fluid, thatthe flow state is laminarflow,, that the temperature of the polymer solution does not change during the flow, and thatthe gravityeffect of gravity onthe solution is not considerednil.
The specific revision content is on the page 6 of the revised article, lines 250-254, the page 8 of the revised article, lines 300-308, and the page 14 of the revised article, line 446.
Point 8: The details about the rheological behavior of the solution in regimes other than "pseudoplastic" are not justified. The power law dependence of viscosity with shear rate is clear in the experiment, so the authors need (should) not propose debatable models for the other regimes.
Response 8: Thank you very much for your valuable suggestions on the study of polymer solution rheological curve. As the polymer solution shows different rheological properties with the increase of shear rate under the action of shear, and consult relevant literature to explain the reasons for the different rheological properties of polymer solution under different shear rates. Referring to your suggestions and the overall contents of the paper, under normal circumstances, the polymer solution shows obvious pseudoplasticity, it is a power-law fluid, and power-law equation is also used as the basis of the rheological model of polymer solution in the partial pressure tools to deduce the rheological equations. Therefore, in order to avoid the misunderstanding of the content of the paper caused by too many rheological models and cause disputes, this paper only considers the rheological equations of the power-law fluid of polymer solution in the pseudoplastic section on the basis of the original polymer solution rheological curve, and does not consider other flow characteristic sections (Newtonian section, Limit Newtonian section, Viscoelastic section and Degradation section.) to make the paper more coherent and the theme and content more explicit.
The specific revision content is on the page 6 of the revised article, lines 237-249.
Point 9: The turbulence of the flow should be distinguished from the presence of a vortex.
Response 9: Thank you very much for your valuable suggestions on the flow state change part of the paper. It is found that the main mechanism of turbulence is the small vortex induced by large vortex through consulting relevant literature. However, the turbulence of the flow is not distinguished from the presence of a vortex in this paper. Therefore, according to your suggestions, this paper supplements and analyzes the phenomenon of flow transition by using turbulence perturbation theory and turbulence dissipation effect.
The specific contents are as follows: “when the polymer solution flows through the throttling section, although the flow direction changes, each layer of fluid maintains its own flow state, does not interfere with the other layers, and maintains a laminar flow state. However, as shown in Figure 6, when the velocity of the polymer solution increases, the fluid layers cannot maintain their respective flow states, the laminar flow state is destroyed, and there is sliding and mixing between the adjacent flow layers, eventually forming turbulent flow. The degree of perturbation of the polymer solution is greatest at the lowest point of the flow channel. By considering the dissipation effect and the turbulent perturbation theory, it is concluded that the turbulence intensity generated by the flow disturbance and the energy loss of the solution are largest in the partial pressure tool.
Figures 7 and 8 are pressure nephograms of the polymer solution flowing in the partial pressure tool under laminar and turbulent states. Under the laminar flow state, the pressure change of the polymer solution is relatively uniform. Under the turbulent flow state, at the place in the tool where the perturbation zone is generated, the pressure suddenly becomes smaller in the upstream of the perturbation zone and becomes larger in the downstream of the perturbation zone, causing an uneven change in the pressure of the polymer solution, which causes the overall pressure in the tool to drop. This is consistent with the analysis of the dissipation effect described above. The development of turbulence accelerates the energy consumption of the polymer solution compared with the laminar flow. The pressure drop is larger than the laminar flow, so it is very important to study the transition from laminar to turbulent flow of the polymer solution in the partial pressure tool.”
The specific revision content is on the page 11 of the revised paper, lines 369-388.
Round 2
Reviewer 1 Report
Accept in present form!
Author Response
Response to Reviewer 1 Comments
Thank you very much for your encouragement of the paper, and thank you for your comments on the paper to make the content and structure of the article more complete and accurate. Finally, thank you very sincerely for your approval and your acceptance of our paper.

Reviewer 2 Report
The authors have improved the manuscript, but I think it is still not suited for publication.
I agree with the authors that polymer flooding uses polymers and is surely very useful, but the science in this manuscript is not about polymers. Only the rheological behavior of the fluid is used. So in my opinon, Polymers is not the best journal for this study.
The introduction has been simplified. It still introduces the context of oil recovery, and it now much more accessible (it could be a bit more concise). The role of the "partial pressure tool" is much better explained: it is used to obtain a large pressure drop while maintaining a high viscosity.
Concerning the experimental curve viscosity vs shear rate, fig 9, I think it supports equation (1) quite well, so it should be moved to the same paragraph.
It is not clear whether figs 10-11 are obtained from the finite element simulations like figs 5-8 or from the rough model. If the finite element simulations are just used for figs 5-8, it should be clearly stated.
Since the partial pressure tool is not generalized, it is worth mentioning from the beginning its typical size (meters or millimeters) even if it is variable. Dimensions are given in the results, but it comes late.
"the turbulence of the flow is not distinguished from the presence of a vortex in this paper."
I think it should be distinguished.
A turbulent flow is chaotic and has a hierarchy of vortex sizes.
The dissipation and pressure drop in the case of a turbulent flow is hard to estimate and is probably very different from the same quantities for a single laminar vortex like observed in the finite element simulation.
Two of the conclusions seem obvious to me, and if they are not, it should be explained why:
1) The pressure drop is higher when the throttling section is smaller. So why not choosing a very small section?
2) The viscosity at the outlet is the same as the viscosity at the inlet (because the geometry, and so the shear rate is the same).
This second point in particular rises a question: If I understood correctly, a high viscosity is desired for the flooding to be more effective. What matters is the viscosity of the polymer solution in the oil layer, not in the partial pressure tool. So what is the idea of controlling the viscosity inside the tool
Author Response
Response to Reviewer 2 Comments
First of all, thank you very much for your comments on this article. We tried our best to improve the manuscript and made some changes in the manuscript. And the replies to your comments are as follows. We appreciate for your warm work earnestly, and hope that the correction will meet with approval. Once again, thank you very much for your comments and suggestions.
Point 1: I agree with the authors that polymer flooding uses polymers and is surely very useful, but the science in this manuscript is not about polymers. Only the rheological behavior of the fluid is used. So in my opinon, Polymers is not the best journal for this study.
Response 1: Thank you very much for your encouragement and support for the content of this paper. The purpose of this paper is to study the rheology of polymer solution in the partial pressure tool for the purpose of improving interlayer contradiction in the process of polymer flooding. Not all fluids are suitable for the research method of this paper, but only polymer solution is taken as the research object, and the consistency coefficient and viscosity index of polymer solution are measured through shear test. The rheological equation and transition velocity of polymer solution in partial pressure tools in this paper are all based on the characteristics of polymer solution, and the rheological properties such as pressure and apparent viscosity are only for polymer solution, other fluids are not applicable. In addition, the paper uses the combination of mathematical model and physical model to study polymer solution, which provides a new method for the future study of polymer solution. Therefore, the research content and method of the article completely conform to the theme of polymer. Therefore, we sincerely hope that you can consider and agree with this paper.
Point 2: The introduction has been simplified. It still introduces the context of oil recovery, and it now much more accessible (it could be a bit more concise). The role of the "partial pressure tool" is much better explained: it is used to obtain a large pressure drop while maintaining a high viscosity.
Response 2: Thank you very much for your comments on the introduction of the paper. According to your comments, we have simplified the introduction of the oil background, kept the content that can explain the problem, and deleted the supplementary content. The specific contents are as follows: “Meyers [4] carried out polymer flooding experiments in two newly developed wells by means of chemically enhanced oil recovery in Kottrabathe oilfield. The results show that the oil recovery efficiency of polymer flooding in the total affected area was 60.6% and the oil recovery efficiency of water flooding in the total affected area was 39.9%. Clark [5] conducted polymer flooding research in the West Kiehl oilfield, and conducted oil displacement tests by means of mobility control, profile modification, and the injection of combined mobility control profile. The experiments showed that polymer flooding successfully displaced residual oil that could not be displaced during water flooding, and the oil recovery ratio increased from 11% to 40%. An oil displacement experiment was carried out by injecting high-concentration polymer solution into heterogeneous sediment cores, Denney [6] concluded from experiments that, when the pressure gradient and interfacial tension of oil layers remain unchanged, polymer flooding can achieve a production efficiency that is 23.87% higher than that of water flooding under the same conditions. Using simulation and core flooding for enhanced oil recovery screening, Wassmuth et al. [7] proposed that, in heavy oil reservoirs, polymer flooding is the preferred enhanced oil recovery technology after primary oil recovery and water flooding. Additionally, when the water cut reaches 95% after water injection, polymer flooding can improve the oil recovery ratio by 20%.”
The specific revision content is on the page 2 of the revised paper, lines 55-70.
Point 3: Concerning the experimental curve viscosity vs shear rate, fig 9, I think it supports equation (1) quite well, so it should be moved to the same paragraph.
Response 3: Thank you very much for your suggestion on the viscosity curve and constitutive equation of polymer solution obtained from shear test of polymer solution. According to your suggestion, in order to better explain the constitutive equation of polymer solution (Equation (1)), we put the relationship diagram between viscosity and shear rate of polymer solution obtained from shear test in the same paragraph as equation (1).
The specific revision content is on the page 6 of the revised paper, lines 238-240.
Point 4: It is not clear whether figs 10-11 are obtained from the finite element simulations like figs 5-8 or from the rough model. If the finite element simulations are just used for figs 5-8, it should be clearly stated.
Response 4: Thank you very much for your suggestions on the influence curves of different structural parameters of partial pressure tools on the pressure and apparent viscosity. Figures 10 and 11 are the pressure and apparent viscosity curve obtained by fitting the rheological equations of the polymer solution in partial pressure tools derived by substituting the consistency coefficient and viscosity index obtained by shear test and different structural parameters of partial pressure tools, which are different from figures 5 and 8 obtained by finite element analysis. The finite element analysis is only used in figures 5 and 8 to illustrate the phenomena of laminar flow and turbulent flow and their effects on pressure, and only to observe abstract concepts that cannot be obtained by calculation. While figures 10 and 11 are used to study the effects of partial pressure tools with different structural parameters on the pressure and apparent viscosity of polymer solution, which can be substituted into relevant parameters and obtained by calculation. On the one hand, the effects of partial pressure tools with different structural parameters on the pressure and viscosity of polymer solution are studied, and on the other hand, the correctness of the deduced rheological equations of polymer solution are verified. The specific content will be supplemented.
The specific contents are as follows: “In order to more clearly observe the change in the flow state of the polymer solution in the partial pressure tool from laminar flow to turbulent flow, as well as the influence of the change in flow state on the pressure of the solution, the Fluent software(ANSYS, USA) was used for numerical simulation. During the simulation process, the model structure was not changed, while the flow rate was gradually increased, thus obtaining a vectorgraph and pressure nephogram of the polymer solution under both a laminar flow and a turbulent flow state. Figure 6 is laminar flow vectorgraph of polymer solution in the partial pressure tool, figure 7 is turbulent flow vectorgraph of polymer solution in the partial pressure tool, figure 8 is laminar flow pressure nephogram of polymer solution in the partial pressure tool, figure 9 is turbulent flow pressure nephogram of polymer solution in the partial pressure tool.”
“The constitutive equation of the polymer solution, the consistency coefficient and rheological index of the shear flow equation obtained by actual fitting, and the structural parameters of the partial pressure tool, are put into the obtained flow characteristic equation, and specific numerical solutions are calculated in order to study the effect of different structural parameters on the pressure drop and viscosity of the polymer solution. Then, the best structural parameters to solve the transition velocity are selected, and the limit value of the flow transition is obtained. The calculated pressure and apparent viscosity of the polymer solution in the partial pressure tools are plotted, as shown in figures 10 and 11.”
The specific revision content is on the page 11 of the revised paper, lines 361-370, and page 14 of the revised paper, lines 447-454.
Point 5: Since the partial pressure tool is not generalized, it is worth mentioning from the beginning its typical size (meters or millimeters) even if it is variable. Dimensions are given in the results, but it comes late.
Response 5: Thank you very much for your suggestions on the size of partial pressure tools. According to your suggestions, we give the typical sizes of partial pressure tools in the annular flow channel and throttling section respectively during the derivation of the rheological equation of polymer solution.
The specific revision content is on the page 7 of the revised paper, line 261, and page 9 of the revised paper, line 313.
Point 6: "the turbulence of the flow is not distinguished from the presence of a vortex in this paper."
I think it should be distinguished.
A turbulent flow is chaotic and has a hierarchy of vortex sizes.
The dissipation and pressure drop in the case of a turbulent flow is hard to estimate and is probably very different from the same quantities for a single laminar vortex like observed in the finite element simulation.
Response 6: Thank you very much for your suggestion to distinguish turbulence from vortex and related problems. The paper is to explain the change of pressure when the polymer solutions in the partial pressure tool are in laminar flow and turbulent flow. The change of pressure in laminar flow is easier to analyze, but the flow in turbulent flow is chaotic, there are a series of vortex sizes, and the dissipation and pressure drop in turbulent flow are difficult to estimate. However, what the paper wants to say is that with the increase of flow velocity, the original laminar flow state cannot be maintained. With the increase of perturbation, laminar flow gradually transits to turbulent flow, and the pressure drop gradually increases in this process. When the turbulent flow state is reached, the pressure drop that can be generated is greater than that of laminar flow state. Therefore, we propose the concept of transition flow velocity, that is, when the flow velocity reaches the full turbulent flow state, we only pay attention to the critical velocity of transition from laminar flow to turbulent flow. And there are theories to judge the critical point from laminar flow to turbulent flow, so the process of transition from laminar flow to turbulent flow is generally ignored in this paper, however, in order to vividly illustrate the difference in physical properties between flow changes, it is used as a comparative test.Therefore, we use Fluent software to analyze the vectorgraph and pressure nephogram of polymer solution in turbulent state by finite element analysis method. In the analysis process, we only pay attention to whether the turbulent state will affect the pressure of polymer solution, and do not consider the numerical value of energy dissipation and pressure drop. From the simulation analysis results, turbulence does produce greater pressure drop, so the concept of transition velocity is proposed in this paper. According to past research, the formula of transition velocity is deduced by the concept of stability factor, ignoring the actual influence of laminar flow and turbulent flow, which helps to simplify the formula derivation. On the other hand, this paper focuses more on the derivation of the transition velocity. In order to highlight the theme of the paper and avoid too much description of laminar flow and turbulent flow phenomena, which do not conform to the theme of polymer, we only observe the phenomena of streamline and pressure change during the transition of polymer solution flow, and make detailed description and derivation of the transition velocity equations of polymer solution, without considering the theoretical analysis of turbulent flow or vortex in detail.
The vectorgraph shown in figure is laminar flow and turbulent flow respectively. For the vectorgraph in turbulent flow, we simulated it by Fluent software. When setting parameters, we selected RNG turbulence model in k-epsilon option, and the velocity at the selected velocity inlet is very large, which can meet the requirement of polymer solution in turbulent flow in the partial pressure tool. As the size of the partial pressure tool is small and the size space of the throttling section is smaller, figures 6 and 7 are enlarged vectorgraphs. In the turbulence vectorgraph, only obvious vortex is observed in the expansion part of the throttling section, and turbulence disturbance is not very obvious due to superposition of vector lines at the contraction section, but actually vortices of different sizes are also generated, and at the outer cylinder wall, due to the small size of the adjacent two parts of the throttling section, the inertia when polymer solution flows, and the streamline overlap is relatively dense, very obvious disturbance is not observed, but at the annular flow channel (i.e., the fully developed section) produces obvious turbulence disturbance and vortices of different sizes. We believe that the flow of polymer solution in the partial pressure tool is turbulent during this process, so the turbulence vector diagram in figure 7 conforms to the turbulent flow state.
Therefore, in summary, in order to highlight the concept of transition velocity of polymer solution and emphasize the derivation of formulas, we only ignore the specific explanation of turbulence phenomenon on the basis of conforming to the actual situation, assisted by finite element analysis, in order to visualize the abstract turbulence phenomenon and provide help for the derivation of the concept and formula of transition velocity.
The fully developed section
Point 7: Two of the conclusions seem obvious to me, and if they are not, it should be explained why:
1) The pressure drop is higher when the throttling section is smaller. So why not choosing a very small section?
2) The viscosity at the outlet is the same as the viscosity at the inlet (because the geometry, and so the shear rate is the same).
This second point in particular rises a question: If I understood correctly, a high viscosity is desired for the flooding to be more effective. What matters is the viscosity of the polymer solution in the oil layer, not in the partial pressure tool. So what is the idea of controlling the viscosity inside the tool
Response 7: Thank you very much for your comments on the conclusion of the paper. Besides the actual production requirements, the selection of the structural parameters of the partial pressure tools should also be based on the difficulties of actual production and cost, combined with practical operation and economic considerations. The shape and structural parameters of the partial pressure tools given in the paper are easier to manufacture and more economical. When selecting a smaller throttling section, although the throttling effect will be better, it is more difficult to manufacture and apply. The higher the pressure-bearing capacity of the material is required, so the oilfield provides the structural parameters of partial pressure tools with outer diameters of 24mm, 24.5mm and 25mm, and front groove angles and rear groove angles of 30°/45°, 45°/30° and 60°/15° for research. In addition, in order to match other tools and installation required by partial pressure injection technology, it is appropriate to select the above parameters. If the selected throttling section area is too small, the molecular chain tensile strength of the polymer solution will increase and the molecular chain will break, which will lead to the failure of the molecular morphology to recover the initial state, the decrease of apparent viscosity and the impact on the oil displacement effect. The area of the selected throttling section should be controlled to a certain size, so the parameters provided can meet the above conditions.
High viscosity oil displacement is indeed an ideal condition for oil displacement, but in the process of oil displacement, high viscosity polymer is easier to be injected into high permeability oil layers, but the smaller the amount injected into thin and poor oil layers, which greatly reduces the permeability and production degree of thin and poor oil layers. Therefore, in the process of polymer flooding, we use a partial pressure tool to control the injection amount of high viscosity polymer in high permeability layers, so that polymer solution can also enter thin and poor oil layers for oil displacement. However, when the partial pressure tool is used for pressure adjustment to control the injection of high viscosity polymer, due to the structural characteristics of the partial pressure tool, the polymer solution will be affected by shearing action when the pressure drops. However, the excessive shearing stress will destroy the polymer molecular chain and cannot restore the original state, resulting in the viscosity drop of the polymer solution and affecting the oil displacement effect of the polymer solution in the high permeability oil layer. In the polymer flooding process, the polymer solution flowing through the partial pressure tool directly enters the formation for oil displacement, and the viscosity of the polymer solution flowing through the partial pressure tool is the viscosity of the polymer solution entering the formation. Therefore, it is extremely important to study the viscosity change of polymer solution in the partial pressure tool. The specific content is already covered in the introduction.
The specific revision content is on the pages 1 to 4 of the revised paper, and page 14 of the revised paper, lines 444-447.
The conclusion part of the paper is summarized based on the structural parameters given in the paper.
The specific revision content is on the page 17 of the revised paper, lines 550-566.
